# Connective tissue fibroblasts from highly regenerative mammals are refractory to ROS-induced cellular senescence

Sandeep Saxena[1], Hemendra Vekaria [2,3], Patrick G. Sullivan [2,3,4] & Ashley W. Seifert [1,3]*

A surveillance system in mammals constantly monitors cell activity to protect against aberrant proliferation in response to damage, injury and oncogenic stress. Here we isolate and culture connective tissue fibroblasts from highly regenerative mammals (*Acomys* and *Oryctolagus*) to determine how these cells interpret signals that normally induce cellular senescence in non-regenerating mammals (*Mus* and *Rattus*). While $H_2O_2$ exposure substantially decreases cell proliferation and increases p53, p21, p16, and p19 in cells from mice and rats, cells from spiny mice and rabbits are highly resistant to $H_2O_2$. Quantifying oxygen consumption and mitochondrial stability, we demonstrate that increased intracellular $H_2O_2$ is rapidly detoxified in regenerating species, but overwhelms antioxidant scavenging in cells from non-regenerative mammals. However, pretreatment with N-acetylcysteine (NAC) protects mouse and rat cells from ROS-induced cellular senescence. Collectively, our results show that intrinsic cellular differences in stress-sensing mechanisms partially explain interspecific variation in regenerative ability.

[1] Department of Biology, University of Kentucky, Lexington, KY 40506, USA. [2] Department of Neuroscience, University of Kentucky, Lexington, KY 40506, USA. [3] The Spinal Cord and Brain Injury Research Center (SCoBIRC), University of Kentucky, Lexington, KY 40506, USA. [4] Lexington VA Medical Center, Lexington, KY 40506, USA. *email: awseifert@uky.edu

ocal cell proliferation is limited during tissue healing in most mammals, with new tissue forming a fibrous scar to repair the injury. During regeneration, resident cells respond to injury by actively proliferating to generate new tissue which subsequently undergoes morphogenesis and growth to replace the damaged or excised part[1]. While there is tremendous interest in understanding the source and developmental potential of the local progenitor cell population in regenerating systems (reviewed in ref. [2]), it is equally important to understand how quiescent cells become activated and transduce injury signals to trigger cell cycle re-entry and controlled cell proliferation[3]. This last question is particularly relevant when trying to understand how closely related species can differ in their injury response; where active cell cycle progression and proliferation occur during regeneration, in contrast to cell cycle stasis and excessive collagen deposition during fibrotic repair[4,5].

Although cell proliferation is tightly regulated during tissue homeostasis in adult vertebrates, a host of stimuli can lead to aberrant cell proliferation and neoplastic transformation of adult cells (reviewed in ref. [6]). Cellular senescence has evolved as an important surveillance mechanism to identify damaged or mutated cells for subsequent removal or isolation. Originally identified to describe the finite proliferative capacity of human cells in culture[7], a range of stimuli are now recognized to induce the stable cell-cycle arrest associated with cellular senescence (reviewed in refs. [8–10]). Among these stimuli, tissue injury causes a local increase in reactive oxygen species (ROS) and can elicit a DNA-damage response (DDR) resulting in stress-induced or damage-induced senescence, respectively. During mammalian tissue repair, proper healing benefits from the secretory phenotype of senescent cells, although efficient clearing of senescent cells is also required for proper tissue remodeling[11–13]. Interestingly, results from several studies suggest that cells in regenerating vertebrates may be refractory to cellular senescence via alternate regulation of tumor suppressor pathways[5,14–16]. This raises the possibility that although injury-propagated cellular senescence might be beneficial for tissue repair in certain contexts, it may also prevent local cell proliferation and thus antagonize regenerative healing[14,16–18].

Comparative models of regenerative success and failure provide an opportunity to investigate if intrinsic cellular differences can explain variation in regenerative ability[19]. Spiny mice (Acomys spp.) are capable of skin and musculoskeletal regeneration, whereas other related murid rodents heal identical injuries via fibrotic repair[5,20–23]. In response to a 4 mm ear punch assay, blastema formation and active cell proliferation distinguish the regenerative response in Acomys from scarring in Mus[5]. While local cells at the injury site were activated to re-enter the cell cycle in both species, significant cell cycle progression and proliferation only occurred during regeneration. Furthermore, injury in Mus induced nuclear localization of the tumor suppressor proteins p21 and p27 during fibrotic repair, whereas these tumor suppressors were not detected in Acomys blastemal cells[5]. In addition to spiny mice, certain lagomorphs can regenerate musculoskeletal tissue and are capable of producing at least 1 cm$^2$ of new skin, cartilage, and connective tissue[5,24–26]. Although cell cycle progression and senescence have not been examined during regeneration in rabbits, alongside spiny mice, they provide an excellent opportunity to ascertain whether key cellular behaviors segregate across healing phenotypes.

In this study, we isolate and culture adult ear pinna fibroblasts from two highly regenerative (Acomys cahirinus and Oryctolagus cuniculus) and two non-regenerating (Mus musculus and Rattus norvegicus) mammals in order to assess proliferative ability and their propensity to experience cellular senescence using a comprehensive panel of consensus senescence markers. For clarity, we refer to these species by their genus throughout the remainder of the paper. First, we show that proliferative ability in culture does not correlate with healing phenotype across species. Next, we find that cells from non-regenerating species rapidly undergo cellular senescence in response to $H_2O_2$ and exhibit strong induction of the consensus markers p21, p53, p16, and p19 (ARF). In stark contrast, cells from regenerating species do not undergo stress-induced senescence in response to $H_2O_2$. While gamma irradiation induced DNA damage and senescence in all species, increased expression of p53 and p21 in Acomys and Oryctolagus was independent of p16 and p19. To mechanistically link ROS and stress-induced senescence we show that increased intracellular $H_2O_2$ is efficiently reduced via glutathione peroxidase (GPx) activity in regenerating species who do not exhibit mitochondrial distress. This contrasts to non-regenerating species which exhibit significant mitochondrial dysfunction in response to $H_2O_2$ exposure. Lastly, although exogenous ROS disrupts mitochondria and triggers cellular senescence in mouse and rat fibroblasts, we demonstrated that pretreatment with NAC protects these cells from ROS-induced cellular senescence.

## Results

**Proliferative ability of ear pinna fibroblasts does not explain healing phenotype.** During vertebrate appendage regeneration, connective tissue fibroblasts are the dominant source for the local proliferative population that will replace the missing tissue[27–29]. To test our hypothesis that connective tissue fibroblasts from regenerating systems exhibit enhanced proliferative ability (compared to cells from non-regenerating animals), we isolated and cultured primary ear pinna fibroblasts from the highly regenerative Acomys and Oryctolagus and from two non-regenerating rodents Mus and Rattus under ambient or physiological oxygen levels (see Methods section). We and others have documented bonafide regeneration in spiny mice in refs. [5,16,20,21,30] and rabbits[5,24,25], and rats have been shown to heal ear punches via fibrotic repair[26]. Under ambient oxygen, Mus fibroblasts entered stasis after ~43 days (mean population doublings (PDs) = 4.4) and Acomys fibroblasts appeared to senesce at ~90 days (PDs = 20.1) (Fig. 1a, b). We hypothesized that Rattus fibroblasts would behave similar to Mus, and Oryctolagus fibroblasts would exhibit enhanced proliferative capacity similar to Acomys. In fact, rabbit fibroblasts exhibited even higher proliferative capacity (107 PDs during 143 days in culture) compared to Mus and Acomys, while rat fibroblasts showed no signs of stasis even after 147 days in culture (PDs = 84.3) (Fig. 1c, d).

Although fibroblasts from some mammals are insensitive to oxygen levels[31], cells from other species (including humans) exhibit impaired proliferative capacity at ambient $O_2$. To better approximate in vivo oxygen concentrations, we assayed cells under physiological $O_2$ levels and asked whether reduced oxygen affected proliferative capacity in adult fibroblasts from Acomys, Mus, Rattus, and Oryctolagus (Fig. 1a–e). We found that physiological $O_2$ significantly increased the proliferative ability of Mus and Acomys cells but had no effect on Rattus and Oryctolagus fibroblasts (Fig. 1a–e). Although Mus cells divided more under reduced oxygen (20.8 ± 0.93 PDs vs. 4.4 ± 0.97 PDs), they still experienced stasis rather quickly (~3 months in culture). Under 3% $O_2$, Acomys fibroblasts divided for ~60 PDs before exhibiting signs of reduced growth and grew at a similar rate to Rattus cells (Fig. 1b, e). In contrast to Mus and Acomys, oxygen concentration did not affect Rattus and Oryctolagus fibroblasts which exhibited almost identical growth rates after 5 months in culture at 20 and 3% $O_2$ (Oryctolagus = 100 PDs, Rattus = 85 PDs) (Fig. 1c, d).

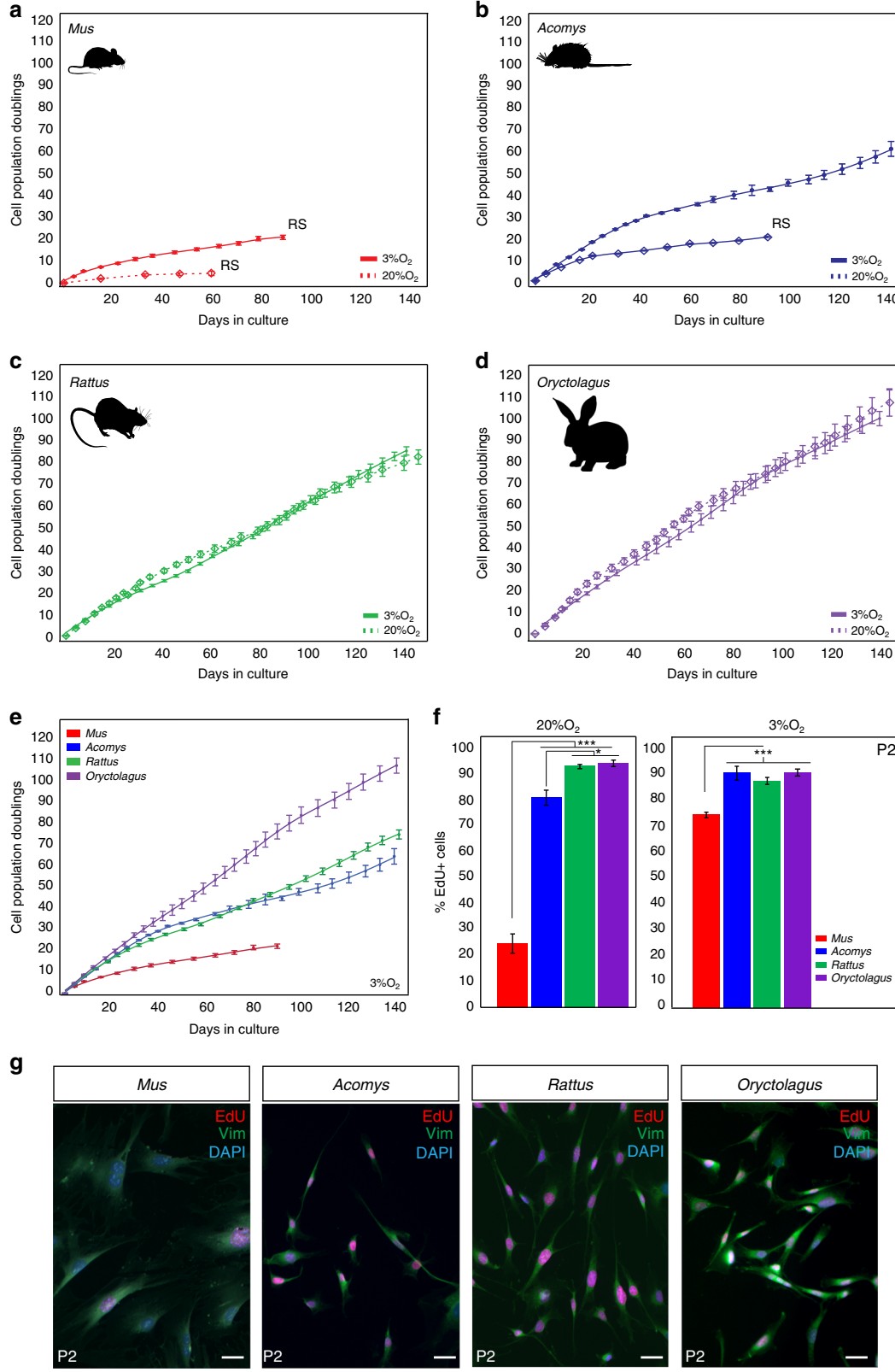

To support these observations, we estimated the proliferative population in culture by labeling passage 2 (P2) cells with EdU for 24 h under ambient and physiological oxygen (Fig. 1f). Under 20% O₂, we found a significantly smaller proliferative population in *Mus* compared to *Acomys, Rattus,* and *Oryctolagus* (ANOVA, $F = 143.39$, $P < 0.0001$) (Fig. 1f and Supplementary Table 1). In addition, the total proliferative population of *Acomys* fibroblasts (82%) was slightly smaller compared to *Rattus* (95%) and *Oryctolagus* (95%) (Tukey-HSD, *Acomys* vs. *Rattus* $t = -4.38$, $P = 0.0043$; *Acomys* vs. *Oryctolagus* $t = -4.87$, $P = 0.0019$) (Fig. 1f). Similarly, at 3% O₂, the percent of EdU + cells was not significantly different between *Acomys* (~88%), *Rattus*

**Fig. 1** Fibroblasts from *Acomys*, *Rattus*, and *Oryctolagus* exhibit enhanced proliferative ability. **a–d** PDs for *Mus*, *Acomys*, *Rattus*,and *Oryctolagus* fibroblasts cultured under ambient (20%) and physiological (3%) $O_2$. **a, b** 3% $O_2$ enhances proliferative capacity of *Mus* and *Acomys* fibroblasts: *Mus* ($n = 5$) cell lines at 3% $O_2$: PDs = 20.8 ± 0.93, vs. 20% $O_2$: PDs = 4.4 ± 0.97 and *Acomys* ($n = 5$) cell lines at 3% $O_2$: PDs = 60 ± 3.4 vs. 20% $O_2$: PDs = 20.1 ± 0.34. **c–d** $O_2$ concentration did not affect mean proliferative ability of *Rattus* ($n = 4$) or *Oryctolagus* ($n = 4$) fibroblasts. **e** Cross-species comparison of proliferative ability at 3% $O_2$ shows that *Mus* cells still senesce while fibroblasts from *Acomys*, *Rattus,* and *Oryctolagus* proliferate for at least 140 days. **f** At 20% $O_2$, the proliferative population (EdU+) of P2 *Mus* cells (~25%) was significantly lower compared to *Acomys* (82%), *Rattus* (95%) and *Oryctolagus* (95%) (ANOVA, $F = 143.3982$, $P < 0.0001$) ($n = 4$/species). Percent EdU + cells in *Acomys* were slightly lower compared to *Rattus* (Tukey-HSD, $t = -4.38$, $P = 0.0043$) and *Oryctolagus* (Tukey-HSD, $t = -4.87$, $P = 0.0019$) ($n = 4$/species). At 3% $O_2$, mean percent EdU + cells in P2 cultures are significantly lower in *Mus* (75%) compared to *Acomys* (88%), *Rattus* (91%) and *Oryctolagus* (91%) (ANOVA, $F = 17.3085$, *Mus* vs. *Acomys*, $P = 0.0003$, *Mus* vs. *Rattus*, $P = 0.0023$ and *Mus* vs. *Oryctolagus*, $P = 0.0002$) ($n = 4$/species). **g** P2 cells co-labeled with EdU and the general fibroblast marker Vimentin demonstrate that > 95% of cell cultures from all four species are fibroblasts ($n = 4$/species). Scale bars = 50 μm. Graphics for *Acomys*, *Mus*, *and Rattus* were made by corresponding author and the *Oryctolagus* image is available free for comercial use. ***$P < 0.0001$, **$P < 0.001$ and *$P < 0.05$. Error bars = S.E.M. Source data are provided as a Source Data file

(~91%), and *Oryctolagus* (~91%), whereas the proliferative rate of *Mus* fibroblasts (~75%) was significantly lower compared to all three species (Fig. 1f and Supplementary Table 2). Alongside EdU, we used vimentin as a broad marker of fibroblast identity and found that our primary cell cultures contained >95% fibroblasts across all four species (Fig. 1g). Collectively, these data show that the intrinsic proliferative ability of ear pinna fibroblasts does not correlate with healing phenotype. In addition, reducing oxygen levels increases the proliferative capacity of *Acomys* and *Mus* cells, but has no effect on the population growth rate of *Rattus* and *Oryctolagus* cells.

**Acomys, Rattus, and Oryctolagus fibroblasts resist senescence in vitro.** We next asked whether enhanced proliferative ability was associated with increased resistance to cellular senescence. To test this association, we assayed progressive passages of fibroblasts from *Acomys*, *Mus, Rattus*, and *Oryctolagus* for low pH β-galactosidase activity (SA-βgal) as a general marker of cellular senescence[32]. Despite an increase in proliferative ability under 3% $O_2$, *Mus* cultures at P3 (PDs = 5.4 ± 0.24) contained 43.6% ± 2.23 SA-βgal + cells and after P13 (PDs = 20.8 ± 0.96) the cultures were completely senescent (Fig. 2a, b). Compared to *Mus* cells at P13, *Acomys* (PDs = 32.5 ± 0.49) and *Rattus* (PDs = 34.0 ± 0.61) exhibited significantly fewer SA-βgal + cells and *Oryctolagus* cultures contained almost no SA-βgal + cells (Fig. 2a and Supplementary Table 3). Multiple stimuli can promote cellular senescence including progressive telomere shortening, DNA damage, de-repression of the cyclin-dependent kinase 2A (*Cdkn2a*) locus, oxidative stress via ROS production and oncogene activation (reviewed in ref. [8]). To more completely identify the senescent phenotype, we used a combination of markers that identify DNA damage/double-stranded breaks (γ-H2AX), *CDKN2A* activation (p19, p16) and general downstream markers of cell cycle inhibition and damage and stress-induced senescence (p21, p53)[8,10]. Because γ-H2AX also labels cycling cells in S-phase, we used EdU to differentiate proliferating cells from non-cycling senescent cells with heterochromatic foci (Fig. 2c). Analyzing P2 cells from *Acomys* and *Mus*, we found significantly more cells (as a percent of total cells in culture) labeled with all of these markers in *Mus* compared to *Acomys* (Fig. 2c–g and Supplementary Table 4). Similar to *Acomys*, and consistent with SA-βgal staining, we observed few percent + cells for these markers in P2 cells from *Rattus* and *Oryctolagus* (Supplementary Fig. 1). These data demonstrate that while *Mus* cells rapidly senesce in vitro, *Acomys*, *Rattus,* and *Oryctolagus* all exhibit prolonged resistance to cellular senescence. These data provide evidence that increased proliferative ability under physiological oxygen concentrations is associated with regenerative ability in some mammals (e.g., *Acomys* and *Oryctolagus*). However, our data for *Rattus* suggested that intrinsic proliferative capacity and resistance to

cellular senescence cannot alone predict enhanced regenerative ability.

**Acomys fibroblasts do not senesce in response to H₂O₂.** Resident fibroblasts within injured tissue experience a number of extrinsic stressors with the potential to drive changes in cellular phenotype. For instance, fibroblasts in the injury environment are exposed to high levels of $H_2O_2$ produced by keratinocytes and inflammatory cells (reviewed in ref. [33]). Thus, we assayed the ability of fibroblasts to withstand $H_2O_2$ to test the in vitro stress response of *Acomys* and *Mus* cells cultured under physiological $O_2$. To quantify stress-induced senescence, we treated cells with 0 μM (control), 75 μM, 150 μM, and 300 μM $H_2O_2$ concentrations for 2 h, removed the $H_2O_2$ and then cultured cells for 24 h and 48 h in fresh media[34]. Twenty-four hours post exposure, we found that the proliferative population in both species was unaffected compared to control samples for all concentrations (Fig. 3a and Supplementary Table 5). However, after 48 h in culture *Mus* fibroblasts exhibited a 37.5% and 59.3% decrease in the percent EdU + cells in response to 150 μM and 300 μM $H_2O_2$, respectively (Tukey-HSD, $t = 4.71$, $P = 0.0019$ and $t = 7.94$, $P = < 0.0001$) (Fig. 3a and Supplementary Table 6). Although the proliferative index of *Acomys* fibroblasts was not significantly altered in response to 150 μM $H_2O_2$ (Tukey-HSD, $t = 1.93$, $P = 0.916$), after 48 h in culture, 300 μM exposure did produce a marginally significant decrease (18.95%) in EdU + cells (Tukey-HSD, $t = 3.32$, $P = 0.0493$) (Fig. 3a and Supplementary Table 6).

Next, we examined how $H_2O_2$ exposure effected cellular senescence. After 24 h in culture, fibroblasts from either species did not exhibit a significant increase in percent SA-βgal + cells in response to any concentration of $H_2O_2$ (Two-way ANOVA, $F = 1.142$, $P = 0.3708$) (Fig. 3b and Supplementary Table 7). In contrast, exposure of *Mus* cells to all sub-lethal concentrations of $H_2O_2$ induced a significant increase in senescence after 48 h in culture (Two-way ANOVA, $F = 223.05$, $P < 0.0001$) (Fig. 3b and Supplementary Table 8). In response to 75 μM or 150 μM $H_2O_2$ exposure, *Acomys* cells did not exhibit an increase in SA-βgal + cells. Although a 300 μM exposure did significantly increase cellular senescence in *Acomys* fibroblasts, the increase was slight (2.14%; Tukey-HSD, $t = -3.75$, $P = 0.0189$) (Fig. 3b and Supplementary Table 8). We also used our panel of senescence markers to examine cellular phenotype in response to $H_2O_2$ exposure (Fig. 3c–e). We found significantly more γ-H2AX + *Mus* cells after only 24 h in culture at all concentrations examined (Fig. 3c and Supplementary Table 9). While this result persisted in *Mus* cells after 48 h in culture, an increase in γ-H2AX + cells was only observed at 300 μM for *Acomys* fibroblasts and only after 48 h in culture (Fig. 3c, d and Supplementary Table 10). Mirroring our results for γ-H2AX, we found that p16, p19, p21, and p53 were significantly increased at all concentrations

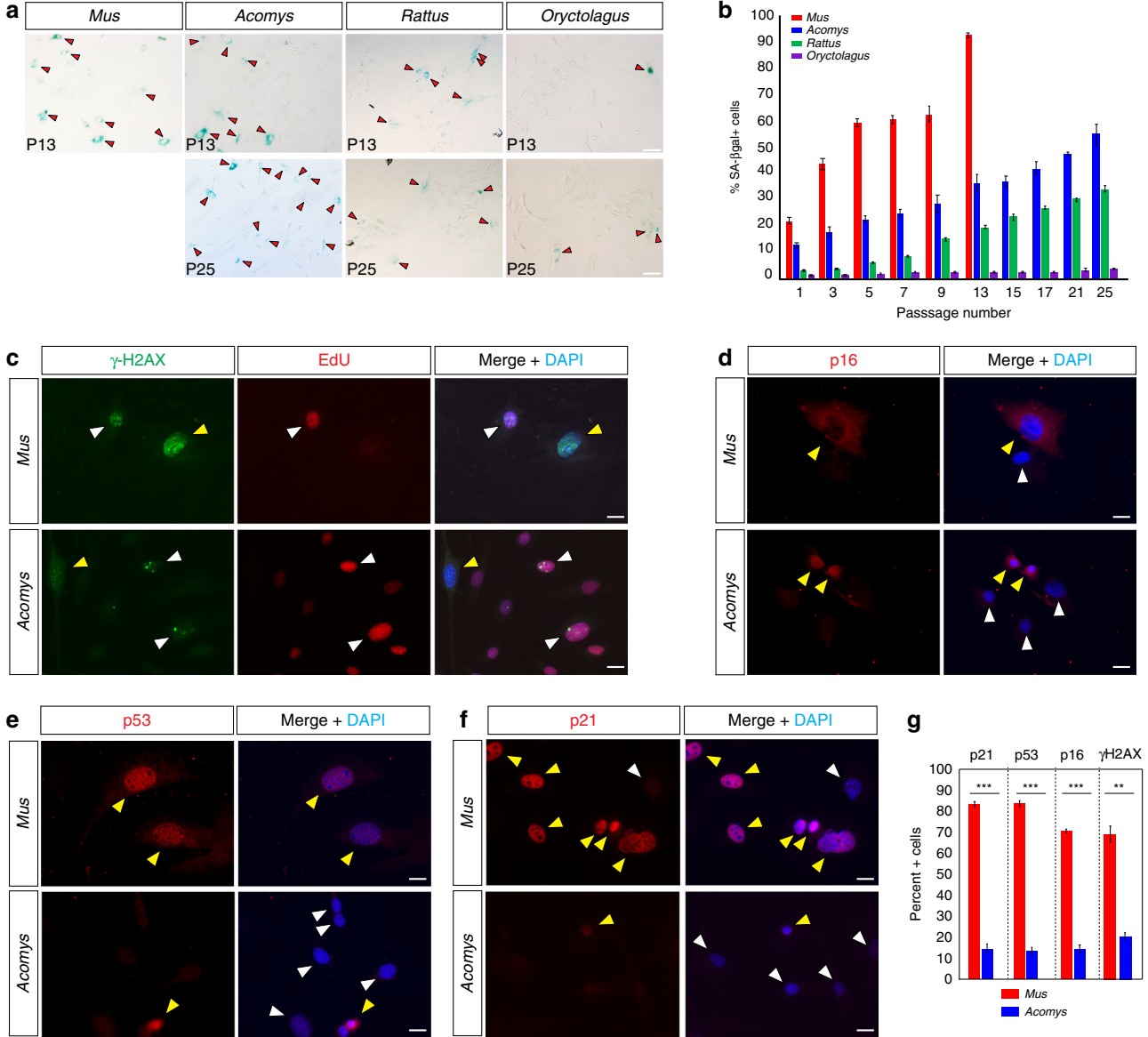

**Fig. 2** In vitro resistance to senescence is not restricted to regenerating mammals. **a**, **b** In line with proliferative ability, cells cultured at 3% $O_2$ were more resistant to cellular senescence. Percent SA-βgal + cells were measured in *Mus*, *Acomys*, *Rattus*, and *Oryctolagus* (*n* = 4–5 species) at progressive passages. *Mus* cells senesced at P13 and > 90% of cultures were SA-βgal+. There were significantly more SA-βgal + cells compared to *Acomys* (~37%), *Rattus* (~22%) and *Oryctolagus* (~4%) (P13 Tukey-HSD, *Mus* vs. *Acomys*, *t* = −23.81, *P* < 0.0001; *Mus* vs. *Rattus*, *t* = 29.29, *P* < 0.0001 *Mus* vs. *Oryctolagus*, *t* = 39.98, *P* < 0.0001). Red arrows indicate SA-βgal + cells (**a**). **c** *Acomys* and *Mus* fibroblasts (*n* = 3/species) from P2 co-labeled with γ-H2AX and EdU to differentiate senescent cells. Yellow arrows indicate nuclei positive for γ-H2AX and EdU and white arrows represent nuclei positive for EdU only. **d–f** P2 *Acomys* and *Mus* fibroblasts (*n* = 3) labeled with p16 (**d**), p53 (**e**), and p21 (**f**), and DAPI. Yellow arrows show marker positive cells while white arrows show negative cells. **g** Quantified cell counts for panels **c–f**. There were significantly more senescent cells in *Mus* cultures positive for: γ-H2AX+, p21+, p53+, and p16+ (Supplementary Table 4). Representative scale bars in panels **a** and **c–f** = 50 μm and 20 μm, respectively. ***P* < 0.0001, ***P* < 0.001, **P* < 0.05. Error bars = S.E.M. Source data are provided as a Source Data file

examined in *Mus* fibroblasts (Fig. 3e and Supplementary Tables 11–14). In contrast, after 48 h in culture we did not detect significant increases for p16, p19, p21, and p53 in *Acomys* cells even after a 300 μM $H_2O_2$ exposure (Fig. 3e and Supplementary Tables 11–14). Taken together, these results show that sublethal concentrations of $H_2O_2$ induce senescence in *Mus* fibroblasts, whereas *Acomys* fibroblasts appear highly resistant to this extrinsic stressor.

**Fibroblasts from regenerating species are refractory to ROS-induced senescence.** Given our finding that *Acomys* fibroblasts appeared to resist ROS-induced cellular senescence, we sought to

test if fibroblasts from *Rattus* and *Oryctolagus* would behave according to their regenerative ability or would instead, reflect their proliferative ability. Repeating our ROS stress experiments with P2 fibroblasts from all four species, we found that $H_2O_2$ elicited a concentration dependent effect on *Mus* and *Rattus* cells (Fig. 4a and Supplementary Tables 15–19). In stark contrast, $H_2O_2$ did not induce senescence in *Oryctolagus* cells at even the highest sub-lethal concentration administered (Fig. 4a). Comparing cells in response to 300 μM $H_2O_2$, we found that the percent of actively proliferating (EdU+) cells significantly decreased after 48 h in *Acomys* (Tukey-HSD, *t* = 4.19, *P* = 0.0099), *Mus* (*t* = 10.04, *P* < 0.0001) and *Rattus* (*t* = 12.88,

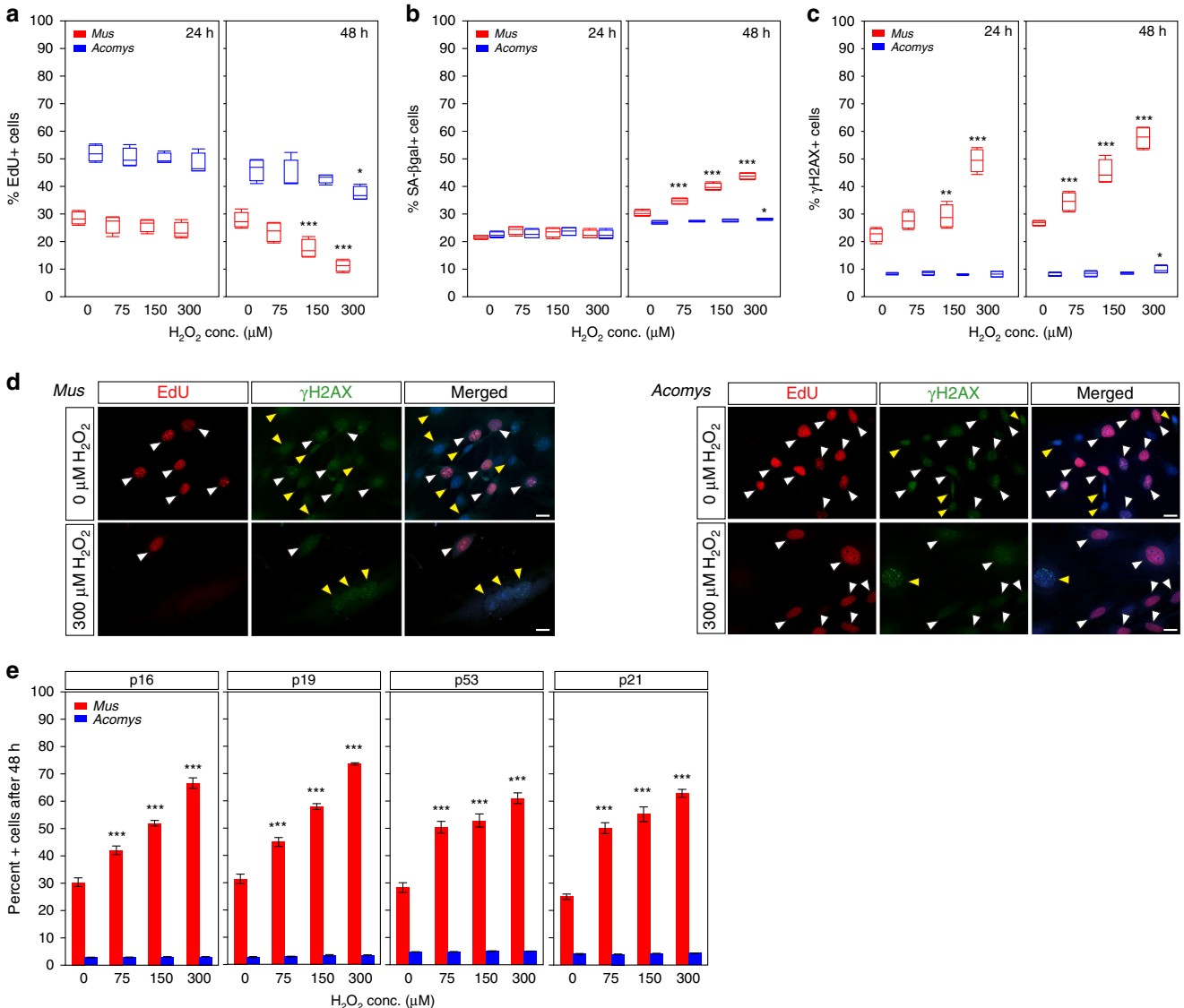

**Fig. 3** $H_2O_2$ exposure does not induce senescence in *Acomys* fibroblasts. **a–e** P2 *Acomys* and *Mus* ($n = 4$/species) fibroblasts treated with sub-lethal doses of $H_2O_2$ (0 μM-control, 75 μM, 150 μM, and 300 μM) for 2 h and then cultured for 24 and 48 h. The horizontal line within the box represents the median sample value of the box plot. The ends of the boxes represent the 25th and 75th quantiles and the lines extending from each end of the boxes are whiskers representing the highest and lowest observations. **a** *Mus* fibroblasts showed no significant change in the percent EdU + cells compared to control after 24 h, but at 48 h experienced a significant decrease in response to 150 μM (Tukey-HSD, $t = 4.71$, $P = 0.0019$) and 300 μM $H_2O_2$ (Tukey-HSD, $t = 7.94$, $P < 0.0001$). No significant changes in *Acomys* EdU + cells at 24 h. A small, but significant decrease in percent EdU + cells after 48 h was detected in response to 300 μM $H_2O_2$ compared to control (Tukey-HSD, $t = 3.32$, $P = 0.0493$). **b** $H_2O_2$ exposure had no effect on cellular senescence in *Mus* and *Acomys* fibroblasts after 24 h in culture. After 48 h in culture, *Mus* fibroblasts exhibited significant increases in SA-βgal + cells at all $H_2O_2$ concentrations, while *Acomys* fibroblasts registered a small, but significant increase at 300 μM only (Tukey-HSD, $t = -3.75$, $P = 0.0189$). **c** *Acomys* fibroblasts followed a similar trend for γ-H2AX + cells after $H_2O_2$ treatment while *Mus* fibroblasts significantly increased γ-H2AX + cells at 24 h in response to 300 μM $H_2O_2$ in addition to significant increases at 48 h in response to all concentrations. **d** Representative cultures for results in panels **a–c** stained with EdU and γ-H2AX. Scale bars = 20 μm. White arrows show double-positive cells and yellow arrows show γ-H2AX + senescent cells. **e** *Acomys* fibroblasts do not upregulate the senescence markers p16, p19, p53, and p21 in response to any concentration of $H_2O_2$. *Mus* fibroblasts show significant increases in these markers compared to *Acomys* in response to all $H_2O_2$ treatments after 48 h (Two-way ANOVA, p16, $F = 783.3681$, $P < 0.0001$; p19, $F = 1094.501$, $P < 0.0001$; p53, $F = 399.1625$, $P < 0.0001$; p21, $F = 526.55$, $P < 0.0001$) (Supplementary Tables 11-14). ***$P < 0.0001$, **$P < 0.001$, *$P < 0.05$. Error bars = S.E.M. Source data are provided as a Source Data file

$P < 0.0001$), while we found no significant change in *Oryctolagus* ($t = 1.09$, $P = 0.991$) (Fig. 4a and Supplementary Table 15). While we found slight, but non-significant increases in *Acomys* and *Oryctolagus* heterochromatic foci, $H_2O_2$ exposure induced large increases in senescent cells in *Mus* and *Rattus* (Fig. 4a, b and Supplementary Table 16). Importantly, we consistently observed that our panel of cellular senescence markers (SA-βgal+,

γ-H2AX+, p21+, p53+, p16+, and p19+) was significantly increased in fibroblasts from the non-regenerating mammals in response to $H_2O_2$ exposure, whereas the regenerating species did not exhibit significant increases in these markers (Fig. 4a, b, Supplementary Fig. 2 and Supplementary Tables 16–21). Together these data show that fibroblasts from regenerating species continue to proliferate and are refractory to ROS-induced

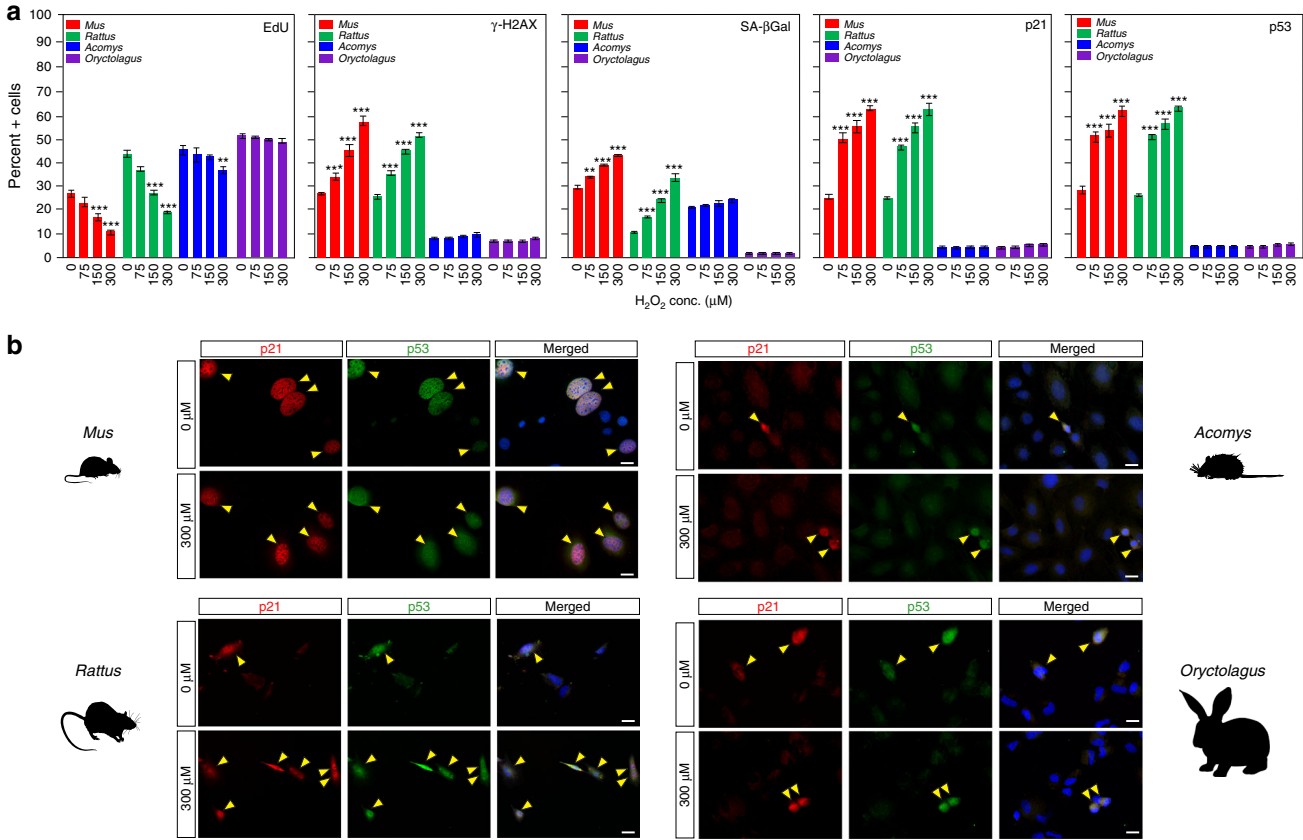

**Fig. 4** Fibroblasts from regenerating mammals are refractory to ROS-induced cellular senescence. **a**, **b** Fibroblasts from regenerating (*Acomys* and *Oryctolagus*) and non-regenerating mammals (*Mus* and *Rattus*) treated with increasing concentrations of $H_2O_2$ (0 μM-control, 75 μM, 150 μM and 300 μM) for 2 h and then cultured for 48 h in complete media (*n* = 4/species). **a** $H_2O_2$ induced a significant decrease in the percentage of proliferating cells (EdU+) in *Mus* at 150 μM (Tukey-HSD, *t* = 5.95, *P* < 0.0001) and 300 μM $H_2O_2$ (Tukey-HSD, *t* = 10.04, *P* < 0.0001), as well as in *Rattus* at 150 μM (Tukey-HSD, *t* = 8.25, *P* < 0.0001) and 300 μM $H_2O_2$ (Tukey-HSD, *t* = 12.88, *P* < 0.0001), a small but significant decrease in *Acomys* at 300 μM $H_2O_2$ (Tukey-HSD, *t* = 4.19, *P* = 0.0099) and no significant change at any $H_2O_2$ concentrations in *Oryctolagus* (Supplementary Table 15). The percentage of senescent cells (γ-H2AX+, SA-βgal+, p21+, and p53+) significantly increased in non-regenerating species and were unchanged in regenerating *Acomys* and *Oryctolagus* (Supplementary Tables 16–19).
**b** Representative fibroblast cultures in panel **a** double-stained with p21 and p53 showing nuclear localization of these tumor suppressors in response to $H_2O_2$. Scale bars = 20 μm. ***P* < 0.0001, **P* < 0.001, *P* < 0.05. Error bars = S.E.M. Source data are provided as a Source Data file

senescence, while fibroblasts from non-regenerating mammals senesce in response to the identical stressor.

**Gamma irradiation induces DDR and cellular senescence**. We next irradiated fibroblasts with increasing amounts of gamma radiation to ascertain if activating a DDR could induce cellular senescence in regenerating species (Fig. 5). Ionizing radiation causes genotoxic damage and activates the DDR, which leads to p53 stabilization and increased levels of p21[35,36]. We used γ-H2AX to monitor DNA damage in response to irradiation[37] and while we observed significantly increased numbers of γ-H2AX+ cells in all four species, the threshold for radiation-induced damage was higher in fibroblasts from *Acomys* and *Oryctolagus* (Fig. 5e and Supplementary Table 22). In addition, increasing amounts of gamma radiation significantly increased p53+ and p21+ cells in a linear fashion among all four species (Fig. 5, Supplementary Fig. 2 and Supplementary Tables 23–24). We also monitored p16 and p19 in response to gamma irradiation and while irradiation led to strongly significant increases in the number of p16+ and p19+ cells even at 5 Gray (Gy) in *Mus* and *Rattus*, irradiation led to very small, albeit significant increases in *Acomys* and *Oryctolagus* at 30 Gy (Fig. 5 and Supplementary Tables 25–26). Together these findings show that fibroblasts from

regenerating species experience senescence in response to irradiation in a p16 and p19 independent manner. In addition, they demonstrate that these tumor suppressor pathways can be activated by extrinsic factors in regenerating species even though they are not activated in response to very high levels of hydrogen peroxide.

**Mitochondria from regenerating species are resilient to $H_2O_2$**. In cells responding to increased oxidative stress, impaired mitochondrial function can lead to cellular senescence[37–39]. We hypothesized that increased intracellular $H_2O_2$ destabilized mitochondria in non-regenerating species. To directly test the effect of $H_2O_2$ exposure on cellular metabolism and mitochondrial function, we quantified the rate of oxygen consumption (OCR) from live cells with and without exposure to 300 μM $H_2O_2$ (Fig. 6a, b). While measuring OCR, we inhibited specific complexes of the mitochondrial respiratory chain using Oligomycin, FCCP, rotenone (+succinate) and antimycin A which allowed us to determine basal respiration, ATP-linked respiration, spare respiratory capacity and maximal respiration (Fig. 6a)[38]. Using this assay, we found that fibroblasts from *Mus* and *Rattus* had impaired mitochondrial function in response to $H_2O_2$ exposure (Fig. 6a, b). This was demonstrated by a significantly decreased

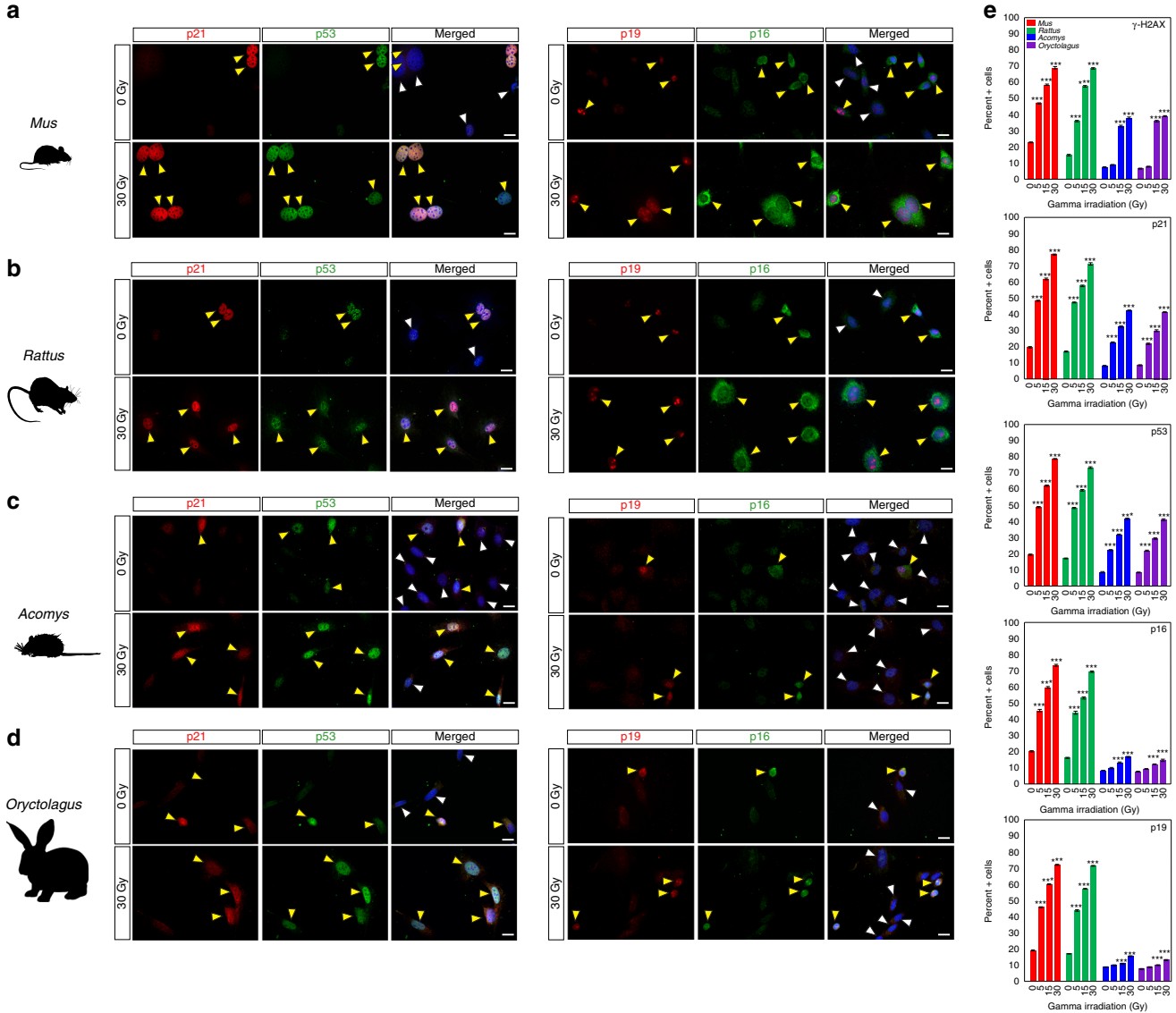

**Fig. 5** Gamma irradiation induces DDR and cellular senescence in all species. **a–d** Cells from all four species were exposed to gamma radiation at 0 (control), 5, 15, and 30 Gy and fixed 6 h later ($n = 4$/species). The representative images show p21, p53, p16, and p19 positive cells in control (0 Gy) and after 30 Gy in *Mus* (**a**), *Rattus* (**b**), *Acomys* (**c**), and *Oryctolagus* (**d**). **e** Fibroblasts from regenerating species showed no significant increase in γ-H2AX + cells at 5 Gy, whereas all species demonstrated a significant increase in DNA damage at 15 and 30 Gy (Supplementary Table 22). Fibroblasts from all four species demonstrated a significant increase in p21+ and p53+ cells at all three radiation doses (Supplementary Tables 23–24). In contrast, fibroblasts from regenerating species showed minimal activation of p16 and p19 in response to gamma radiation, where fibroblasts from *Mus* and *Rattus* strongly activated p16 and p19 (Supplementary Tables 25–26). Scale bars in panels **a–d** = 20 μm. ***$P < 0.0001$, **$P < 0.001$, *$P < 0.05$. Error bars = S.E.M. Source data are provided as a Source Data file

percent-normalized OCR, whereas no significant changes were observed in fibroblasts from regenerating species (Fig. 6b). In addition, all the components of mitochondrial respiration we quantified were significantly decreased in response to $H_2O_2$ treatment in non-regenerators (Fig. 6a, b and Supplementary Tables 27–30). In stark contrast, *Acomys* and *Oryctolagus* exhibited no mitochondrial dysfunction in response to $H_2O_2$ exposure (Fig. 6a, b and Supplementary Tables 27–30).

We next asked whether increased intracellular $H_2O_2$ directly impacted mitochondrial function across species. To do this, we isolated mitochondria 2 h after treating fibroblasts with 0 μM (control) or 300 μM $H_2O_2$ and performed assessments across different states of respiration using ADP + pyruvate/malate, oligomycin, FCCP and Antimycin A. This combination of inhibitors assessed bioenergetic flux and the intactness of isolated

mitochondria. The OCR for isolated mitochondria after providing excessive ADP and pyruvate/malate, yields the state III respiration rate. We found a significant decrease in state III respiration (ATP production) for mitochondria from *Mus* and *Rattus* (ANOVA, *Mus*: $t = 2.39$, $P = 0.0292$ and *Rattus*: $t = 2.85$, $P = 0.0116$), whereas we observed no change in state III respiration for *Acomys* and *Oryctolagus* (ANOVA, *Acomys*: $t = 0.04$, $P = 0.9676$ and *Oryctolagus*: $t = 0.00$, $P = 0.9996$) (Fig. 6c and Supplementary Table 31). We next looked at the intactness of mitochondria by assessing the coupling of respiration and phosphorylation. As ADP is exhausted, oligomycin blocks complex V and inhibits ADP to ATP conversion thus decreasing the OCR and providing the state IV respiration rate, a measure of proton leak across the inner mitochondrial membrane. Thus, the respiratory control rate (RCR) (the ratio of state III/IV respiration

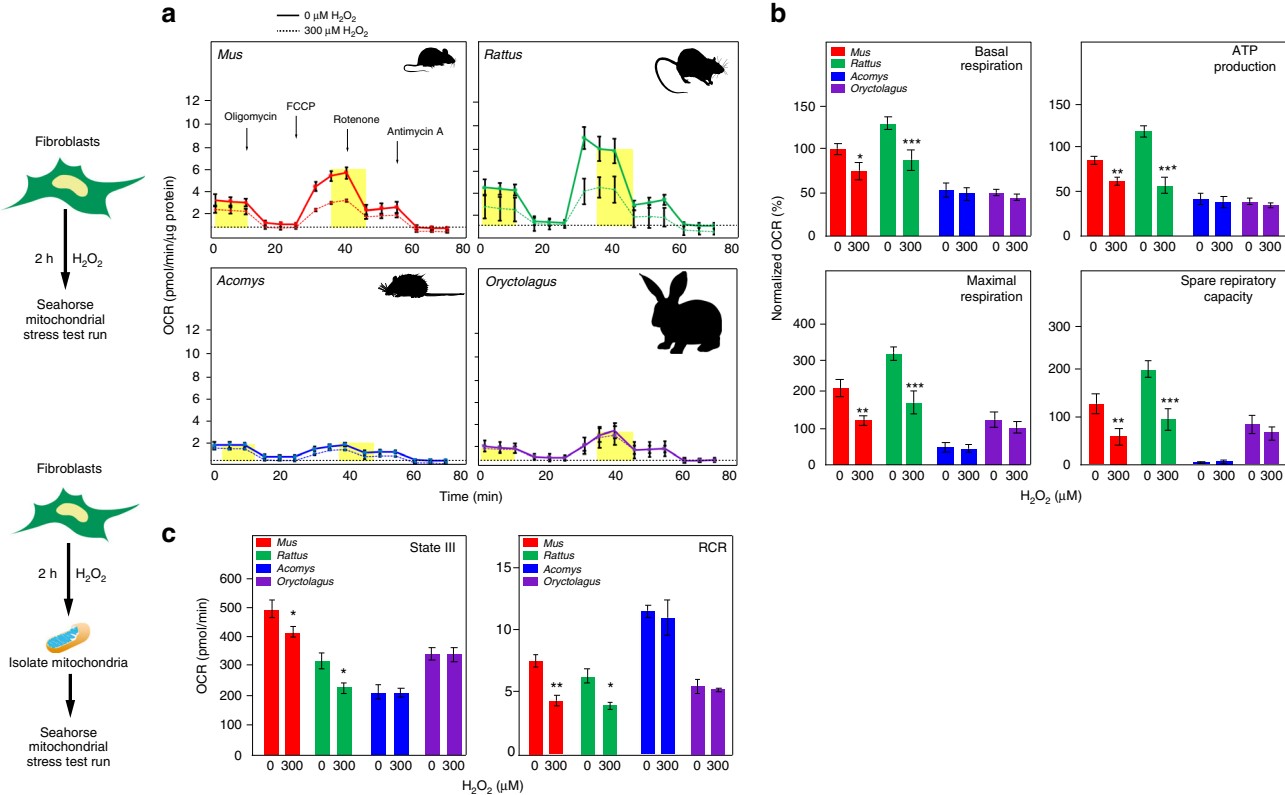

**Fig. 6** Mitochondria from regenerating species are resilient to $H_2O_2$. **a** Mitochondria stress testing measured as OCR (pmol/min/μg protein) for fibroblasts from all species after control (PBS) and 300 μM $H_2O_2$ treatment. Mitochondrial complex inhibitors were used to generate measures for basal respiration, maximal respiration, ATP production, and spare respiratory capacity. **b** Measured OCR was converted to percent normalized OCR and we found that all measured parameters were significantly decreased in fibroblasts from non-regenerating species: basal respiration (ANOVA, $F = 15.8779$, $P < 0.0001$), ATP production (ANOVA, $F = 25.2168$, $P < 0.0001$), maximal respiration (ANOVA, $F = 20.2582$, $P < 0.0001$), and spare respiratory capacity (ANOVA, $F = 15.0920$, $P < 0.0001$) ($n = 5$/species). **c** The mitochondria were isolated from all the species after 0 μM $H_2O_2$ (control) or 300 μM $H_2O_2$ treatment ($n = 3$/species). Mitochondria assays were performed on a XF96 analyzer using ADP + pyruvate/malate and mitochondrial inhibitors oligomycin, FCCP and Antimycin A. State III respiration was achieved through ADP stimulated respiration whereas State IV respiration was calculated after oligomycin inhibition. The Respiratory Control Rate (RCR) was calculated as the ratio of State III and State IV respiratory rate. The OCR for State III and RCR was significantly decreased after $H_2O_2$ treatment in purified mitochondria of non-regenerators while no significant effect was shown by regenerators: State III respiration (ANOVA, $F = 19.1281$, $P < 0.0001$) and RCR (ANOVA, $F = 19.1558$, $P < 0.0001$). Graphics for fibroblast and mitochondria were adapted from open source material. ***$P < 0.0001$, **$P < 0.001$, *$P < 0.05$. Error bars = S.E.M. Source data are provided as a Source Data file

rate) provides a measure of how well respiration and phosphorylation are coupled. Analyzing the RCR, we found that it too was significantly decreased in mitochondria from *Mus* and *Rattus* after $H_2O_2$ treatment (ANOVA, *Mus*: $t = 3.41$, $P = 0.0035$ and *Rattus*: $t = 2.55$, $P = 0.0215$) (Fig. 6c and Supplementary Table 32). Again, and in contrast to the non-regenerating mammals, we found no significant change in the RCR for mitochondria from *Acomys* and *Oryctolagus* (ANOVA, *Acomys*: $t = 0.53$, $P = 0.6005$ and *Oryctolagus*: $t = 0.28$, $P = 0.7856$) (Fig. 6c and Supplementary Table 32). In combination with our assessment of intact cells, our results demonstrate that mitochondria from regenerating species are somehow protected from exogenous ROS (or insensitive to it). In contrast, $H_2O_2$ exposure destabilizes mitochondria and reduces cellular metabolic rate in fibroblasts from non-regenerating mammals. These data also suggest that mitochondrial dysfunction in response to injury stress may be a source of increased ROS production and senescence in non-regenerators compared to regenerators.

**Acomys and Oryctolagus rapidly detoxify exogenous $H_2O_2$.** To address the potential for enhanced cytoprotection in fibroblasts from *Acomys* and *Oryctolagus*, we measured intracellular $H_2O_2$

levels using genetic (HyPer) and chemical (PO1) sensors pre-$H_2O_2$ and post-$H_2O_2$ treatment (Fig. 7a, b and Supplementary Fig. 3). HyPer is a highly sensitive, genetically encoded fluorescent probe that can specifically detect $H_2O_2$ in cells[39]. First, we transfected fibroblasts from all four species with HyPer, passaged cells in culture and then transferred cells to an incubator mounted on an inverted microscope where equal numbers of cells were cultured for 48 h. This allowed us to monitor, measure, and track fluorescent intensities of individual cells over time (Fig. 7a, b). Prior to $H_2O_2$ treatment, we selected groups of transfected cells for live-imaging every 30 min (see Methods). Treatment wells were administered 300 μM $H_2O_2$ and cells within control and treatment wells were imaged over 18 h (Fig. 7a, b). Cells showed negligible movement during imaging and after imaging, we chose twenty cells per cell line for all four species ($n = 3$ lines/species) that exhibited similar fluorescence measures prior to treatment and used those cells to calculate fluorescent intensity ratios for progressive time points (Fig. 7b). These conditions allowed us to monitor a single wavelength to quantify intracellular $H_2O_2$ levels[39–42]. This data showed peak fluorescence levels were not different across species 30 min after $H_2O_2$ treatment (Fig. 7b and Supplementary Tables 33, 34). However, whereas fluorescence levels rapidly returned to baseline after 2.5 h in

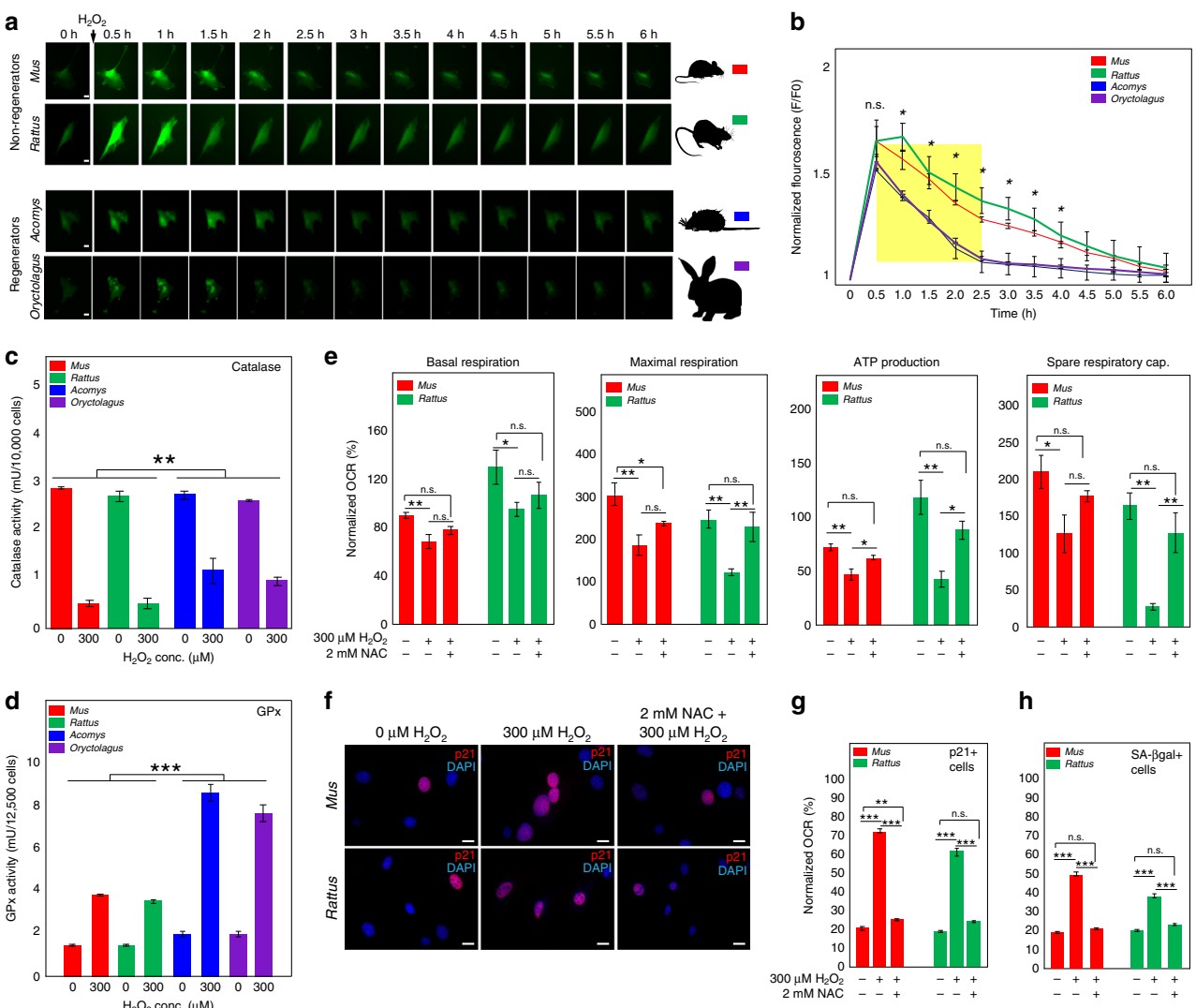

**Fig. 7** Regenerators rapidly detoxify exogenous hydrogen peroxide via increased GPx activity. **a** Representative images for a single live-imaged cell transfected with HyPer which captured every 30 min over 6 h. **b** Fluorescence intensity ratios (F/F0) were calculated for >20 cells/cell lines for all four species ($n = 3$ lines/species). A linear model was used to detect $H_2O_2$ detoxification rate and ANOVA was used to detect significant differences between species. F/F0 quantified 30 min after $H_2O_2$ treatment were not significantly different between species (ANOVA, $F = 1.2153$, $P < 0.3651$ and Supplementary Table 34). $H_2O_2$ was detoxified significantly faster in *Acomys* and *Oryctolagus* (yellow box) (Linear model, $F = 3.9693$, $P < 0.0137$ and Supplementary Tables 33, 34). After 30 min, F/F0 were significantly different between species until 4 h post treatment (Supplementary Table 34). In *Acomys* and *Oryctolagus*, F/F0 returned to baseline levels after ~2.5 h (Tukey-HSD, *Acomys*: $t = −1.42$, $P = 0.1592$ and *Oryctolagus*: $t = −1.70$, $P = 0.0929$), whereas it did not return to baseline in *Mus* and *Rattus* until ~5 h post-treatment (Tukey-HSD, *Mus*: $t = −1.74$, $P = 0.0853$ and *Rattus*: $t = −1.91$, $P = 0.0588$). Asterisk refers to significant differences between species from an ANOVA. **c** Catalase activity was significantly decreased in all species following $H_2O_2$ exposure (Supplementary Table 35). Comparing catalase activity between regenerating and non-regenerating species revealed a significantly greater decrease in non-regenerators (Two-way ANOVA, LS Means, $F = 20.5504$, $P = 0.0003$). **d** GPx activity was significantly increased in response to 300 μM $H_2O_2$ in all species (Supplementary Table 36). However, GPx activity increased to a greater extent in regenerators compared to non-regenerators (Two-way ANOVA, LS Means-, $F = 172.5647$, $P < 0.0001$). **e** Pre-treatment with NAC protected *Mus* and *Rattus* cells from mitochondrial destabilization upon $H_2O_2$ exposure. Percent normalized OCRs were not significantly different among non-treated and NAC + $H_2O_2$ (pre-treatment with 2 mM NAC followed by 2 h of 300 μM $H_2O_2$ treatment) treated samples in *Mus* and *Rattus* for all the measured components: basal respiration, maximal respiration, ATP production and spare respiratory capacity (Supplementary Tables 37–40). **f–h** Pre-treatment with NAC prevents ROS-induced senescence in *Mus* and *Rattus* cells as measured by percent p21 + (ANOVA, *Mus*, $F = 1309.3470$, $P < 0.0071$ and *Rattus*, $F = 258.1558$, $P < 0.0924$) and SA-βgal + cells (ANOVA, *Mus*, $F = 669.5451$, $P < 0.1465$ and *Rattus*, $F = 125.3335$, $P < 0.0522$) staining. Representative scale bars in panel **a** = 50 μm and in panel **f** = 20 μm. ***$P < 0.0001$, **$P < 0.001$, *$P < 0.05$. Error bars = S.E.M. Source data are provided as a Source Data file

*Acomys* (Tukey-HSD, $t = −1.42$, $P = 0.1592$) and *Oryctolagus* (Tukey-HSD, $t = −1.70$, $P = 0.0929$), fluorescence levels remained significantly higher in *Mus* (Tukey-HSD, $t = −3.06$, $P = 0.0028$) and *Rattus* (Tukey-HSD, $t = −3.63$, $P = 0.0004$) till at least 4 h post treatment (Fig. 7b and Supplementary Table 34). $H_2O_2$ levels returned near baseline after 5 h in *Mus* (Tukey-HSD,

$t = −1.74$, $P = 0.0853$) and *Rattus* (Tukey-HSD, $t = −1.91$, $P = 0.0588$) (Fig. 7b and Supplementary Table 34). Monitoring $H_2O_2$ levels pre-treatment and post-treatment using the chemical probe PO1 revealed a similar pattern for each species and showed that intracellular $H_2O_2$ is more rapidly degraded in cells from regenerating mammals (Supplementary Fig. 3). These data

demonstrate that cells from all four species experience the same acute intracellular $H_2O_2$ burst. However, rapid ROS degradation in *Acomys* and *Oryctolagus* shorten intracellular exposure to $H_2O_2$ compared to *Mus* and *Rattus* cells.

Could rapid detoxification of $H_2O_2$ in fibroblasts from regenerating mammals be explained through higher antioxidant activity? Among ubiquitous antioxidant scavengers, catalase and GPx protect cells from oxidative stress. Catalase converts $H_2O_2$ into water and oxygen, and GPx decomposes $H_2O_2$ into water. We assayed the activity of both enzymes from fibroblasts at baseline and following a 1 h exposure to $H_2O_2$ (Fig. 7c, d). While baseline catalase activity was not significantly different among species (*Mus*: $2.8 \pm 0.01$ mU/ml, *Rattus*: $2.7 \pm 0.09$ mU/ml, *Acomys*: $2.7 \pm 0.08$ mU/ml, *Oryctolagus*: $2.6 \pm 0.02$ mU/ml) (Two-way ANOVA, $F = 97.7947$, $P < 0.0001$), in response to a 1 h $H_2O_2$ exposure, catalase activity decreased twice as much in non-regenerating species (*Mus*: $0.5 \pm 0.06$ mU/ml, *Rattus*: $0.5 \pm 0.09$ mU/ml, *Acomys*: $1.2 \pm 0.24$ mU/ml, *Oryctolagus*: $1 \pm 0.08$ mU/ml) (Two-way ANOVA, LS Means-, $F = 20.5504$, $P < 0.0003$) (Fig. 7c and Supplementary Table 35). These small differences in catalase activity suggested it was not responsible for the rapid conversion of $H_2O_2$ in cells from regenerating species. Similar to endogenous catalase activity, baseline GPx activity was not significantly different across species (*Mus*: $1.49 \pm 0.01$ mU/ml, *Rattus*: $1.46 \pm 0.04$ mU/ml, *Acomys*: $1.93 \pm 0.12$ mU/ml and *Oryctolagus*: $1.95 \pm 0.14$ mU/ml) (Two-way ANOVA, $F = 172.3622$, $P < 0.0001$) (Fig. 7d and Supplementary Table 36). However, after a 1 h $H_2O_2$ treatment, GPx activity was significantly elevated in fibroblasts from *Acomys* and *Oryctolagus* compared to fibroblasts from the non-regenerating species (*Acomys*: $8.53 \pm 0.38$ mU/ml and *Oryctolagus*: $7.55 \pm 0.42$ mU/ml, *Mus*: $3.79 \pm 0.01$ mU/ml and *Rattus*: $3.48 \pm 0.07$ mU/ml) (Two-way ANOVA, LS Means-, $F = 172.5647$, $P < 0.0001$) (Fig. 7d and Supplementary Table 36). This activity difference mirrored the comparatively lower levels of intracellular $H_2O_2$ we observed in non-regenerating species using HyPer and PO1 fluorescence (Fig. 7a, b and Supplementary Fig. 3). These data reveal a general pattern across all four species where catalase levels were quickly overwhelmed by $H_2O_2$ treatment and GPx activity increased to protect cells from oxidative stress. Moreover, our findings suggest that GPx could more rapidly detoxify exogenous $H_2O_2$ in fibroblasts from regenerating species and in doing so enhance cytoprotection from intracellular ROS damage.

**NAC protects *Mus* and *Rattus* cells from ROS-induced senescence.** Protein alignments for the three major mammalian GPx proteins (i.e., GPx1-3) revealed the highly conserved nature of these enzymes and suggested that enzyme structure alone was unlikely to drive differences in activity across healing phenotype (Supplementary Fig. 4). Instead, GPx relies on the bioavailability of glutathione (GSH) to reduce $H_2O_2$. We reasoned that increasing GPx activity in *Mus* and *Rattus* fibroblasts might protect these cells from ROS-induced cellular senescence. NAC is the N-acetyl derivative of cysteine and is a general glutathione precursor[43]. Moreover, administering NAC effectively increases GSH stores thereby increasing GPx activity[44]. We conducted dose-response studies and determined that pretreatment with 2 mM NAC for 1 h effectively increased antioxidant scavenging in the presence or absence of $H_2O_2$ (Supplementary Fig. 5a, b). Next, we pre-treated cells for 1 h with NAC, added 300 μM $H_2O_2$ to treatment wells (or nothing in control wells), washed cells 2 h later with PBS and performed mitochondrial stress testing on live cells (Fig. 7e). Using this experimental design, we found that NAC treatment protected cells from the deleterious effects of exogenous ROS (Fig. 7e). While $H_2O_2$ treatment led to significant

decreases in basal respiration, maximal respiration, ATP production and space respiratory capacity, NAC treatment significantly protected cells from the negative effects of $H_2O_2$ exposure (basal respiration: ANOVA, *Mus*, $F = 5.9922$, $P < 0.0157$; *Rattus*, $F = 2.0781$, $P < 0.1070$; maximal respiration: ANOVA, *Mus*, $F = 8.5244$, $P < 0.0050$ and *Rattus*, $F = 7.3260$, $P < 0.0073$; ATP production: ANOVA, *Mus*, $F = 10.5917$, $P < 0.0022$ and *Rattus*, $F = 11.1963$, $P < 0.0018$; Spare respiratory capacity: ANOVA, *Mus*, $F = 4.2258$, $P < 0.0408$ and *Rattus*, $F = 14.4430$, $P < 0.0006$) (Supplementary Tables 37-40). We then asked whether increased GPx activity could similarly protect *Mus* and *Rattus* cells from ROS-induced cellular senescence. Indeed, pre-treatment of fibroblasts with 2 mM NAC effectively abolished significant increases in p21 + and SA-βgal + cells in response to $H_2O_2$ exposure (Fig. 7f–h and Supplementary Tables 41-42). Taken together, our results show that increased GPx activity correlates with increased cytoprotection in *Acomys* and *Oryctolagus* to ROS-induced cellular senescence. Furthermore, our data support that increasing cellular detoxification of $H_2O_2$ via NAC can protect *Mus* and *Rattus* fibroblasts from mitochondrial dysfunction and ROS-induced cellular senescence.

## Discussion

During epimorphic regeneration, local cells proliferate after being triggered to undergo cell cycle re-entry and progression. In contrast, injury signals during fibrotic repair trigger local cell activation, but cells undergo cell cycle arrest. The mechanistic difference between this differential response to injury remains unknown. One possibility is that injury signals induce local cells to senesce which prevents them from undergoing cell cycle progression[14]. While senescent cells and the mechanisms that generate them have been well-studied in the context of aging and cancer, how (or if) senescent cells contribute to complex tissue regeneration has only recently begun to attract attention. Senescent cells occur during regeneration and fibrotic repair, but they account for a relatively small fraction of cells within the injury microenvironment during regeneration[45]. During salamander limb regeneration, p53 activity is reduced in blastemal cells along with the p53 targets *gadd45* and *mdm2* but returns to baseline during cell differentiation suggesting a role in regulating cell proliferation[46]. In contrast to mammalian myotubes, serum stimulation induces newt myotubes to readily undergo cell cycle progression and injecting newt myotubes with human p16 blocks cell cycle re-entry[15]. Human p19 (ARF) can similarly inhibit fin regeneration when inserted into zebrafish under a heat shock promoter where activation following injury negatively regulates blastema cell proliferation[16]. Mammalian myotubes are normally resistant to cell cycle re-entry and progression but can be induced to re-enter the cell cycle following knockdown of pRb and p19 (ARF)[14]. Together, these studies suggest that cells from regenerating and non-regenerating vertebrates possess important differences in cellular surveillance that ultimately regulates proliferative ability, although this idea has not been tested in closely related species who can and cannot regenerate.

Previous work from our group showed that a major difference between regeneration or fibrotic repair in *Mus* and *Acomys*, respectively, was a lack of local cell proliferation in *Mus*[5]. This disparity in proliferation was accompanied by resident *Mus* fibroblasts shuttling p21 and p27 to the nucleus, whereas during regeneration, p21 and p27 were restricted proximal to the injury site. To extend these findings, we turned to analyzing ear pinna fibroblasts from *Mus* and *Acomys* in vitro where we could pinpoint intrinsic cellular mechanisms that could help explain our in vivo findings. Our comparative analysis of fibroblasts from *Acomys* and *Mus* showed that *Acomys* fibroblasts exhibited

significantly greater proliferative ability and were highly resistant to stress-induced cellular senescence mediated by $H_2O_2$, a prominent ROS found in regenerative and non-regenerative injury microenvironments[18,30,33,47]. Broadening our analysis to include ear pinna fibroblasts from another regenerating (*Oryctolagus*) and non-regenerating (*Rattus*) mammal, we demonstrated that enhanced proliferative ability was not explicitly restricted to regenerating species as *Rattus* fibroblasts proliferated well in culture. However, when exposed to $H_2O_2$ and gamma radiation, *Rattus* and *Mus* fibroblasts rapidly senesced with induced expression of γ-H2AX, SA-βgal, p21, p53, p16, and p19. In contrast, *Acomys* and *Oryctolagus* fibroblasts were refractory to cellular senescence in response to oxidative stress. Gamma irradiation did, however, induce significantly increased expression of senescent markers in the regenerating species, although few cells exhibited p16 and p19. Thus, while proliferative capacity appeared to be species-specific, the cellular response to ROS was strictly regeneration-associated. Coupled with our in vivo data showing an absence of these senescence-associated proteins in the blastema during regeneration in *Acomys*[5], our in vitro data suggests local, injury-induced ROS may activate local cells in both healing contexts, but that sustained ROS induces senescence only in non-regenerating species. These results move beyond ROS as an affecter of cell behavior during regeneration and instead implicate a species-specific response to ROS as a key component for how local cells respond to injury.

Although poorly understood at a mechanistic level, ROS are required for tail regeneration in larval *Xenopus*[48,49] and zebrafish[50]. In addition, $H_2O_2$ produced in response to ventricular heart resection is required for regeneration[51]. $H_2O_2$ is generated immediately upon injury over a 100–200 μm margin from the amputation site[47,51] which covers the proximal zone from which blastemal cells are generated[1,52]. While the early burst of $H_2O_2$ is an important chemotactic signal for leukocytes, it also stimulates local cell proliferation and selective inhibition of $H_2O_2$ production using diphenyleneiodonium (DPI) inhibits blastemal or myocardial cell proliferation[18,51]. That early exposure to $H_2O_2$ mediates a proliferative response which occurs days later suggests that ROS signals may play an early and pivotal role in activating progenitor cells. Indeed, recent work showed that hydrogen peroxide and superoxide are generated locally during regeneration and fibrotic repair in *Acomys* and *Mus*, respectively[30]. Moreover, prolonged superoxide production from NADPH-oxidase was significantly higher during regeneration compared to fibrotic repair, and this exposure may serve to facilitate cell proliferation after the first wave of acute inflammation subsides[30]. The influx of inflammatory cells and concomitant release of ROS into the injury microenvironment are shared components of the wound healing phase of regeneration and fibrotic repair, a phase that is more similar than different between healing modalities. Our results suggest that cellular sensitivity to extracellular stressors, rather than the comparative magnitude of the inflammatory response, drives regeneration or scarring. Furthermore, our data suggests that the early ROS burst associated with acute inflammation may set local cells along two different response pathways through the differential effect of ROS on cellular homeostasis.

Mitochondrial stability is integral to cellular homeostasis and disruption of mitochondrial function has been shown to contribute to disease pathogenesis[53,54]. Although mitochondrial integrity has rarely been examined in the context of tissue regeneration[55], it seems reasonable to consider that mitochondria might represent an important cellular sensor to exogenous stressors in the injury microenvironment. In an attempt to determine how ROS-stress induced cellular senescence in fibroblasts, we examined mitochondrial function across species at baseline and in response to exogenous ROS. Our data revealed a direct correlation between the stress-induced senescent phenotype in cells from non-regenerating species and mitochondrial response. This included a significant decline in basal respiration, ATP production from oxidative phosphorylation, maximal respiration rate and spare capacity in response to hydrogen peroxide exposure. Surprisingly, mitochondria from regenerating species exhibited no signs of mitochondrial dysfunction despite elevated levels of intracellular $H_2O_2$.

Our data provide support for two mechanisms limiting stress-induced senescence via mitochondrial breakdown. First, our finding that hydrogen peroxide exposure significantly increased glutathione peroxidase activity in fibroblasts from regenerating species (compared to non-regenerating species) shows that anti-oxidant scavengers can be mobilized to limit acute stress. Importantly, our experiments using NAC to boost glutathione levels and increase GPx activity in fibroblasts from *Mus* and *Rattus* showed that ROS-induced cellular senescence could be prevented by mimicking a mechanism deployed by fibroblasts from regenerating mammals. This demonstrates that altering antioxidant scavenging can enhance cytoprotective mechanisms that are in place during regeneration and this holds promise for altering healing outcomes in non-regenerating mammals. These findings also open the door to future experiments aimed at determining the kinetics of GSH metabolism/glutathione stores in regenerating versus non-regenerating mammals. Second, we observed lower baseline OCR in regenerating species suggesting that mitochondria from regenerating species may be less reliant on ATP production from oxidative phosphorylation which in turn could increase the resistance threshold for mitochondria in response to stress or damage. Together, our results demonstrate that increased intracellular $H_2O_2$ overwhelms compensatory mechanisms that protect regenerating species, and that subsequent mitochondrial disruption triggers cellular senescence. Importantly, our results raise the possibility that $H_2O_2$ or other early ROS signals may work synergistically with other signaling molecules to activate and facilitate cell cycle progression in regenerating species. While extrapolating in vitro results to the dynamic wound environment is not without its caveats, our in vivo and in vitro data implicate the cellular response to ROS as a key component of regenerative healing.

## Methods

**Animals**. African spiny mice (*Acomys cahirinus*) and outbred laboratory mice (*Mus musculus*-ND4, Envigo) were housed in our animal housing facility at the University of Kentucky, Lexington, KY[56]. *Mus* were maintained on 18% mouse chow (Tekland Global 2918, Harlan Laboratories, Indianapolis, IN) and *Acomys* were maintained on 14% mouse chow (Teklad Global 2014, Harlan Laboratories) and black sunflower seeds. Animals were exposed to natural light through large windows (approximately 12:12 h L:D light cycle during the experiment). New Zealand white rabbits (*Oryctolagus cuniculus*) and rats (*Rattus norvegicus*) were maintained by the Division of Laboratory Animal Research, University of Kentucky. Ear pinna fibroblasts were acquired from sexually mature *Acomys* (12-week-old to 20-weeks old), *Mus* (10-week-old to 14-week-old), *Oryctolagus* (16-week-old to 28-week-old) and *Rattus* (16-week-old to 28-week-old). Animal work was approved by the University of Kentucky Institutional Animal Care and Use Committee (IACUC) under protocol 2013–1119.

**Fibroblast acquisition and culture**. Connective tissue fibroblasts from all four species were acquired from uninjured ear pinna. Briefly, for each experimental animal, a 4 mm ear punch biopsy was used to remove a full-thickness piece through the center of the ear pinna. Tissue was pooled from both ears of a single individual and single cell suspensions were made by enzymatic (Trypsin/Dispase) and mechanical digestion (chopping with blade/passing through 70 μm filter). Cells were cultured in complete-DMEM media (10% FBS (Hyclone), 1% L-glutamate (Hyclone), and 1% Antibiotic-Antimycotic (Gibco) and grown at 37 °C in either ~20% oxygen and 5% $CO_2$ or 3% oxygen and 5% $CO_2$. Cells were passaged at ~80% confluency and counted using a hemocytometer. After every passage ~200,000 cells were seeded in a 10 cm culture plate (Cyto-one). Passage number and doubling times were calculated as previously published[57] using the formula:

Population Doublings (PDs) = log [(number of cells harvested)/(number of cells seeded)]/log2

**SA-βgal staining**. SA-βgal staining was used as a general marker of cellular senescence[32,58]. Twenty thousand cells/well were cultured for 3 days in 6-well plates. After three days in culture, cells were fixed with 2%v/v formaldehyde and 0.2%v/v glutaraldehyde in PBS for 15 min and later, washed 3 × (5 min each) with PBS. The fixed cells were stained with SA-βgal staining solution at pH 6.0 (40 mM citric acid/phosphate buffer pH-6, 150 mM NaCl, 5 mM $K_3Fe(CN)_6$, 5 mM $K_4Fe$ $(CN)_6$, 2 mM $MgCl_2$, and 1 mg/ml X-gal in water) and incubated in the dark at 37 °C incubator for 48 h. Stained cells were washed with deionized water. The SA-βgal stained cells were visualized under bright field microscopy, photographed, and counted.

**Immunocytochemistry**. For all immunocytochemistry, ~5000 cells/well were grown on cover slips in 24-well plates for 48 h. After 48 h, cells were fixed for 10 min with 10% formalin and washed 3× with PBS (5 min/wash). Fixed cells were permeabilized for 15 min with PBT (0.5% Triton X-100 in PBS) and blocked with 1.5% donkey serum in PBS for 30 min. Cells were subsequently incubated overnight with specific primary antibody at 4 °C (see below). Following incubation with primary antibody, cells were washed with PBS, incubated with the appropriate species-specific secondary antibody for 30 min at room temperature, washed in PBS, and nuclei stained with DAPI (Invitrogen). Cover slips were mounted on slides with Prolong Gold antifade reagent (Invitrogen). Negative controls were run using isotype specific antibodies at the appropriate concentration. For EdU staining, growing cells were treated with EdU (2 μM) for 24 h. Cells were fixed for 10 min with 10% formalin and washed thrice with PBS. EdU labeling was performed using click chemistry where Cu (I) induced a reaction between Azide 594 (0.5 mg/2 ml, Life Technologies) and the alkyne present in EdU. DAPI (blue) was used to stain nuclei. In cases where multiple labeling was performed, the EdU reaction was performed before antibody staining. Primary antibodies used were: p16 (p16[INK4A]) at 1:100 (rabbit polyAb, 10883-I-AP, Proteintech), p19 (p19[ARF]) at 1:100 (rat mAb, sc32748, Santacruz), p21 at 1:1 (rat mAb, HUGO 321 C/C5, Monoclonal antibodies core unit, Spanish National Cancer Research Center), p53 at 1:500 (rabbit pAb, NCL-P53-CM5P Leica/Novacastra), γH2AX at 1:500 (rabbit mAb, P-H2A.X-Ser139, clone 20E3, Cell signaling technology) and Vimentin at 1:100 (rabbit mAb, 5741 S, Cell signaling technology) (Supplementary Fig. 6a–f). Secondary antibodies used were: Donkey anti-rat Alexa flour 488 (A21206, 2 mg/ml, Invitrogen, 1:500) and Donkey anti-rabbit Alexa flour 594 (A21207, Life technology, 2 mg/ml, 1:500). To quantify positive cells, ~10 images were captured at ×20 magnification using an Olympus BX53 microscope with each image containing 20–25 cells such that >250 cells were counted per sample. Cells positive for individual markers were counted manually and calculated as a percent of all cells present (DAPI + cells).

**Hydrogen peroxide exposure and gamma irradiation**. P2 cells from *Acomys*, *Mus*, *Rattus*, and *Oryctolagus* were cultured on cover slips in 24-well plates (~5000 cells/well) for 48 h after which they were either treated with sub-lethal doses of $H_2O_2$[34] or transiently exposed to sub-lethal doses of gamma radiation. Cells were treated for 2 h with $H_2O_2$ at either 0 μM (control), 75 μM, 150 μM and 300 μM concentrations. After 2 h cells were washed in PBS, fresh media added and cultured for either 24 or 48 h. The cells were treated with 10 μM EdU for 3 h before fixing at 24 or 48 h. The treated cells were fixed with 10% formalin for 10 min and washed 3× with PBS. After 48 h, cells were gamma irradiated at a dose rate of 1.66 Gy/min using a Mark I-68 137Cesium γ-irradiator set to 0 Gy (control), 5 Gy, 15 Gy, or 30 Gy and treated cells were cultured for an additional 6 h. Cells were treated with 10 μM EdU for 3 h before fixation. Fixed cells from both treatments were prepared for immunocytochemistry as outlined above.

**Intracellular hydrogen peroxide detection**. To assess intracellular $H_2O_2$ we used HyPer, a highly sensitive, genetically encoded fluorescent probe that can specifically detect $H_2O_2$ in cells[39] and PO1. HyPer is a genetically encoded fluorescent sensor which has yellow fluorescent protein (cpYFP) attached to the two regulatory domains of OxyR[39]. The interaction of HyPer with $H_2O_2$ leads to formation of intracellular disulfide bond that creates conformational change, as well as change in the fluorescent intensity of cpYFP. Intracellular $H_2O_2$ levels are usually quantified by calculating the ratio of two excitation wavelengths, 420 nm and 500 nm. However, it is appropriate to use single wavelength monitoring if cell movement is minimal and sensor expression levels are similar across cells as occurs in our experimental design[39–42]. For experiments using HyPer, equal numbers of cells were seeded into 24-well glass bottom plates until they reached 70–80% confluency. Cells from each well were then transfected with 0.5 μg the HyPer plasmid (pHyPer-cyto, FP941, Evrogen) using Lipofectamine 3000 transfection reagent (L3000001, Thermo Fisher Scientific) according to the manufacturer's protocol and conditions optimized for each species. We prepared plasmid DNA (Opti-MEM medium, P3000 reagent and 0.5 μg Hyper plasmid) and lipid (Opti-MEM medium and lipofectamine 3000 reagent) complexes and incubated for 15 min. Plasmid DNA-lipid complexes were added to each well and incubated for 48 h before treatment and live imaging.

PO1 is a fluorescent probe that readily diffuses across the cell membrane and reacts chemoselectively with hydrogen peroxide using a boronate oxidation trigger[59]. Upon reaction with free $H_2O_2$, an arylboronate group is converted to phenol which produces a fluorescent product. An equal number of fibroblasts (~5000 cells/well) from *Acomys*, *Mus*, *Rattus*, and *Oryctolagus* were cultured for 48 h on cover slips in 24-well plates. Cells were assayed using PO1 after being treated continuously for either 30, 60, 90, or 120 min with 300 μM $H_2O_2$. In addition, after a continuous 120 min exposure to 300 μM $H_2O_2$ we also assayed cells at either 30 min, 6, 12, 24, and 48 h post treatment. After each individual treatment duration, cells were washed twice with PBS and treated with 10 μM PO1 (CAS Number 1199576-10-7, Tocris biosciences). PO1 exposed cells were then washed thrice with PBS and fixed for 10 min using 10% formalin. The cells were washed again with PBS and stained with DAPI. DAPI-stained cells were washed and mounted using Prolong gold. In order to photograph cells, we calculated the time of peak PO1 intensity for each species and subsequently photographed all time points using this signal intensity.

**Microscopy and image acquisition**. Bright field images for SA-βgal stained cells were taken using an Olympus IX-71 with DP72 color camera (12.8 megapixel cooled digital color camera). Fluorescent images were captured using a BX53 microscope (Olympus, Tokyo, Japan) with DP80 Color Camera (Dual CCD Color and Monochrome Camera) and equipped with cellSens software (cellSens v1.12, Olympus Corporation).

Live imaging of HyPer transfected cells was performed for 18 h using an Olympus IX83 inverted, fluorescent microscope outfitted with an automatic stage and culture chamber. Culture conditions in the chamber were maintained as above and with physiological oxygen levels. The fluorescent intensity of HyPer was measured using a FITC channel with excitation and emission wavelength (ex/em) at 495/520 nm. Although the HyPer probe is often detected by calculating the ratio of two excitation wavelengths (420 and 500 nm), we monitored a single wavelength because there was negligible movement of fibroblasts during live imaging and we controlled for sensor transfection from all four species[39–42]. To account for potential variation in sensor transfection, we analyzed cells (≥20 cells/cell line) exhibiting similar baseline expression levels of HyPer across all four species. Different fields of fluorescent cells were imaged using ×20 objective lens in each well prior to $H_2O_2$ treatment. Cell fields were tagged using cellSens software for automated image capture every 30 min for 18 h. Captured images were analyzed using Olympus cellSens software by measuring the fluorescent intensity of each cell (≥20 cells/cell line) at each time point. The fluorescent intensity at each time point for a single cell was divided by the pre-treatment intensity of the same cell to calculate the fluorescent intensity ratio (F/F0). Ratios were calculated from three independent cells lines for each species.

**Catalase and GPx activity assays**. To measure catalase and GPx activities we used enzyme assay kits (Abcam; Catalase-ab83464 and GPx-ab219926). Fibroblasts from *Acomys*, *Mus*, *Rattus*, and *Oryctolagus* were cultured in 10 cm petri dishes until 90% confluency. Cells were counted, and cell lysates were prepared from an equal number of cells (500,000 cells/100 μl) according to the manufacturer's protocol. For the catalase activity assay, we treated cell lysates with 0 μM and 300 μM $H_2O_2$ for 1 h and then measured non-decomposed $H_2O_2$ using an oxired probe to produce a fluorescent product at 535/587 nm (Ex/Em). Hence, catalase activity is inversely proportional to the measured signal. For the GPx assay we obtained lysates from ~12,500 cells. This assay relies on several reactions. The first reaction involves conversion of reduced GSH to oxidized glutathione (GSSG). Later this GSSG and NADPH chemically interact to produce GSH again and NADP + with the help of glutathione reductase (GR). In this GPx activity assay, we used a NADP sensor which only interacts with NADP+ to produce a fluorescent product at 420/480 nm (Ex/Em). Hence the GPx activity is directly proportional to the signal obtained.

**Mitochondrial stress test**. To test the effect of $H_2O_2$ treatment on mitochondrial function, we performed mitochondria stress tests using a Seahorse XFe96 Analyzer (Agilent). We measured OCR in live cells and in response to specific inhibitors of the mitochondrial respiratory complexes: Oligomycin, FCCP, rotenone (+succinate) and antimycin A. Oligomycin inhibits ATP synthase and reveals ATP-linked respiration, FCCP uncouples oxygen intake and ATP production and reveals maximal respiration, and rotenone blocks complex I and antimycin A inhibits complex III disrupting all mitochondrial respiration. Using this design, we quantified basal respiration, ATP-linked respiration, proton leak-linked respiration, spare capacity, maximal respiration capacity, and non-mitochondrial respiration of the cultured cells. An equal number of fibroblasts (~40,000 cells/well) from *Acomys*, *Mus*, *Rattus*, and *Oryctolagus* were cultured for 24 h in Seahorse XF cell culture 96 well-microplates. After 24 h, we treated fibroblasts from all four species with 0 μM $H_2O_2$ or 300 μM $H_2O_2$ for 2 h and washed with PBS. We prepared DMEM Xf assay media (with 1 mM pyruvate and 5 mM glucose) and adjusted the pH to 7.4. The treated cells were incubated in this specialized media at 37 °C in a non-$CO_2$ incubator for 1 hr. We hydrated the sensor cartridge in Seahorse XF Calibrant overnight at 37 °C in a non-$CO_2$ incubator. We injected the sensor cartridge ports with 1.5 μM oligomycin in port

A, 2 μM FCCP in port B, 0.8 μM rotenone in port C and 1 μM antimycin A in port D. We loaded a calibration plate along with a sensor cartridge plate to calibrate the Seahorse Xf-96 analyzer. Following calibration, we inserted our treated cell cultured plate and ran the capture program. The program includes five different cycles with three OCR measurements each for baseline followed by injection of oligomycin, FCCP, rotenone and antimycin A, respectively. The measured OCR was normalized with BCA quantified protein present in the sample. To avoid variation in multiple runs, this normalized OCR was normalized again with the basal value of non-treated *Mus* and we calculated the percentage for each sample in every run.

**Mitochondrial isolation**. Isolated mitochondria were purified using ultra-centrifugation and Ficoll gradients[60,61]. Cells from *Acomys, Mus, Rattus,* and *Oryctolagus* were cultured in 10 cm petri dishes until 90% confluency and then treated with 0 μM and 300 μM $H_2O_2$ for 2 h. After 2 h, the cells were washed in HBSS, trypsinized, counted, and the cell pellet kept on ice. The cell pellet was resuspended in 1 ml of mitochondrial isolation buffer and centrifuged. 400 μl fresh mitochondrial isolation buffer was added and resuspended. The samples were kept in an $N_2$ bombardment chamber at 1300–1400 psi pressure for 10 min at 4 ℃. These samples were passed four times through a 22-gauge syringe. The samples were centrifuged at low speed (1.3 rcf) for 3 min at 4 ℃ and the supernatant was transferred into a fresh tube. These tubes were then centrifuged at 13,000 rpm for 10 min at 4 ℃ and discarded the supernatant. The cell pellets were resuspended in 400 μl fresh mitochondrial isolation buffer and placed atop a discontinuous Ficoll gradient (10% and 7.5%) and ultracentrifuged at 32,000 rpm for 30 min at 4 ℃. The purified mitochondrial pellets were resuspended into fresh mitochondrial isolation buffer and centrifuged at 13,000 rpm for 10 min at 4 ℃. The supernatant was discarded and the pellets were resuspended in fresh mitochondrial isolation buffer. The resuspended samples were quantified using a BSA assay and then 8 μg/sample was used for a Seahorse run. Mitochondrial OCR were measured in respiration buffer using Seahorse flux analyzer (Agilent) with serial injections of (Pyruvate + Malate + Adenosine diphosphate (ADP)) via port A to induce State III (measures ATP production); Oligomycin via port B to induce State IV (measures proton leak across inner mitochondrial membrane); FCCP via port C to induce State $V_{CI}$ and (Rotenone + Succinate) via port D to induce State $V_{CII}$ modes of respiration (measures maximum electron transport system capacity thru complex I and complex II, respectively).

**N-acetyl cysteine treatment**. N-acetyl Cysteine (NAC) was acquired from Sigma-Aldrich (A7250). An equal number of *Mus* and *Rattus* cells (40,000 cells/well) were seeded separately on two different 96-well Seahorse plates and cultured for 24 h. Next, cells from both plates were pre-treated with different concentrations (0 μM, 500 μM, 1 mM, 2 mM, 5 mM, and 10 mM) for either 1 h or 6 h prior to $H_2O_2$ treatment to determine the optimal NAC dosage across species. Each NAC treatment was followed by either 0 μM or 300 μM $H_2O_2$ treatment for 2 h. After 2 h, cells were washed with HBSS and fresh media was added. Next, we performed a mitochondrial stress test (as outlined above) using a Seahorse XFe96 Analyzer. After completing the mitochondrial stress test, cells were lysed in each well and total protein was quantified for every well using a BSA assay to normalize the Seahorse run. These normalized OCR's from two plates were again normalized to a basal value of their respective non-treated samples (non-treated *Mus* sample for *Mus* plate and non-treated *Rattus* sample for *Rattus* plate). The percentage was calculated for each sample in every run. The optimal dose across species was determined to be a 1 h pretreatment with 2 mM NAC.

*Mus* and *Rattus* cells were cultured on 24-well plates with coverslips. The cells were treated with 2 mM NAC for 1 h followed by $H_2O_2$ for 2 h. After 2 h, cells were washed with HBSS and fresh media was added. Cells were fixed after 48 h. Staining for p21 and SA-βgal was performed as described above.

**Statistical analyses**. JMP (version Pro 12.1.0, SAS Institute Inc.) was used to perform all statistical analyses. In all tests, alpha was set at 0.05. Percentage data was arcsine transformed for analyses and expressed graphically as percent. Post hoc comparisons were made using Tukey-HSD multiple comparisons. To analyze percent SA-βgal + cells (as a fraction of total cell number) between *Acomys* and *Mus* we used a two-way ANOVA with species, passage number and the species\*passage number interaction followed by post-hoc comparisons to test for the effect of species and passage number. To analyze our immunocytochemical data for different senescent markers (represented as a percentage of total cells) between *Acomys* and *Mus*, we used a two-way ANOVA with species, senescent marker and the species\*cell marker interaction followed by post-hoc analysis to test for the effect of species and cell marker. To identify significant proliferative differences (percent EdU + cells) among the four species we used a one-way ANOVA. In order to analyze percent SA-βgal + cells (as a fraction of total cell number) and test for significant differences between species and passage number, we used a repeated measures two-way ANOVA with species, passage and the species\*passage interaction as factors. Post-hoc comparisons tested for an effect of species and passage number. To analyze data generated for the $H_2O_2$ exposure experiments, we quantified percent proliferating (EdU+) and percent positive

cells (i.e., SA-βgal + , γ-H2AX, p16, p19, p21, and p53) and used a two-way ANOVA to test for an effect of species, $H_2O_2$ concentration and the species\*$H_2O_2$ concentration interaction. We used Tukey HSD to specifically test for significant differences within species. We used a linear model (from 0.5 h to 2.5 h) to compare the differences in fluorescent intensities ratio (F/F0) after $H_2O_2$ treatment on HyPer transfected cells of P2 *Acomys, Mus, Rattus,* and *Oryctolagus*. We performed ANOVA followed by *t*-test to compare fluorescent intensities ratio at 0.5 h among all the 4 species. We also analyzed the significant changes in fluorescent intensities ratio after 2.5 h, 4 h and 5 h among all the 4 species through ANOVA followed by *t* test multiple comparisons. We analyzed the catalase and GPx activity assays by using a two-way ANOVA and multiple comparison post hoc test for the effect of species, $H_2O_2$ concentration, and species\*$H_2O_2$ interaction. We also used a LS Means test to compare the changes in catalase and GPx activity among the regenerating and non-regenerating species groups. Similarly, to analyze the mitochondrial OCR data for basal respiration, ATP production, maximal respiration and spare respiratory capacity, we used a two-way ANOVA and multiple comparison *t*-test for the effect of species, $H_2O_2$ concentration and the species\*$H_2O_2$ interaction. We analyzed the isolated mitochondrial OCR data for State III and RCR using two-way ANOVA and multiple comparison *t*-test for the effect of species, $H_2O_2$ concentration and the species\*$H_2O_2$ interaction. Further, we used One-way ANOVA to analyze OCR data of pretreated NAC samples followed by $H_2O_2$ treatment for basal respiration, ATP production, maximal respiration and spare respiratory capacity, as well as for percent p21+ and SA-βgal+ cells. Relevant statistics are reported in the results and figure legends. Complete statistical outputs are reported in Supplementary Tables 1–42.

## Data availability
All relevant data that support the findings contained within this study are embedded in the paper and Supplementary Information. Numerical data are contained in the source data file and any additional data are available from the corresponding author upon reasonable request.

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

## Acknowledgements

We thank all members of the Seifert lab for insightful discussions, Thomas Gawriluk and Shishir Biswas for help with comparative genomics, Fatemah Safaee for help with routine cell culture, Adam Cook for animal husbandry, Xiang Liang for help with irradiation experiments, Katherine Thompson, Arnold Stromberg, and Aviv Brockman for consultation on statistical analysis. This work was supported by grants from the National Science Foundation (NSF) and the Office for International Science and Engineering (OISE) (IOS-1353713) and from the National Institute of Musculoskeletal, Arthritis, and Skin Diseases (NIAMS) (R01AR070313) to A.W.S. The content in this article is solely the responsibility of the authors and does not necessarily represent the official views of the National Institutes of Health or National Science Foundation.

## Author contributions

S.S. and A.W.S designed the project and experiments. S.S. performed all the experiments with significant help from H.V. and P.S. in analyzing mitochondrial function. S.S. and A.W.S analyzed the data and S.S. and A.W.S. wrote the paper with input from P.S. All authors commented on and edited the final version.

## Competing interests

The authors declare no competing interests.
