## [Peer Review File · Nature Communications]

Reviewers' Comments:

Reviewer #1:

Remarks to the Author:

This ms. by Seifert and colleagues investigates the basis for differential regenerative capacity between different mammals (using cultured cells as a model system), focusing on differences (and underlying basis) between propensity to undergo senescence.

Comparing senescence, proliferative capacity between different regenerative and non-regenerative they find inhibition of senescence in cells from regenerative mammals, but proliferative capacity does not correlate with regenerative capacity. Further investigating the mechanistic basis for regenerative differences the authors describe increased stress resistance to oxidative stress (H₂O₂) induced senescence, accompanied by increased levels of catalase and an insensitivity of mitochondria to oxidative stress.

This study is undoubtedly interesting with potentially important implications. However, the study suffers from being wholly descriptive - its unclear whether many of the effects shown here (e.g. catalase upregulation/oxidative stress/response of mitochondria to oxidative stress) are relevant for the differences in regenerative capacity since no experiments have been done to directly address this. Key points the authors may want to consider are the following.

- the authors show nicely that cells from regenerative animals are resistant to oxidative stress induced senescence (through addition of H₂O₂) - how relevant this is for the regenerative difference between cells/organism is unclear. Is there increased oxidative stress in the cell lines reported, and (more importantly) does manipulation of endogenous oxidative stress levels affect regenerative capacity? e.g. does ROS scavenging significantly impede senescence in cells for non-regenerative animals

- in the same line of determining cause and effect, Figure 6 title states "Increased GPx activity and mitochondrial resilience protect regenerating species from stress-induced senescence." - this is not what is shown in figure 6, merely that there is a correlation. Does manipulating catalase activity impact on senescence in this setting (and, more importantly, under endogenous conditions?)

Reviewer #2:

Remarks to the Author:

In this manuscript, Saxena et al. address some of the mechanistic differences between fibroblasts from regenerative versus related non-regenerative mammals. As the authors identify in references (e.g. Bryant et al. 1986), fibroblasts likely play an important role in the regenerative capacity of a tissue and therefore mechanistic insight on their cellular differences, correlated with that capacity, is an important area of research. By using freshly isolated fibroblast cells, the authors aimed to exclude the variables in regenerative capacity due to tissue environment in the different mammals and focus on the intrinsic properties of the fibroblasts. To this end, the authors began by measuring the senescence and proliferation rates of cultured cells as a function of time in the typical models of the lab mouse and rat, and also the regenerative models of the African spiny mouse and the rabbit ear. The authors used culture conditions (3% oxygen) that improved the proliferative capacity of the mouse and rat cells, and which led them to investigate the effect of increased oxygen stress (H₂O₂), as well as gamma radiation, on the fibroblasts. Finally, the authors identify a mechanistic difference between the regenerative and non-regenerative cells (glutathione peroxidase), which can account for the observed different responses to the oxidative stress.

The authors show some very interesting findings on the increased resistance to oxidative damage

and gamma radiation damage in the fibroblasts which segregate neatly between regenerative and non-regenerative species. However, the main conclusion that the authors draw from Figures 1 to 3 about proliferative capacity and regenerative potential seem ambiguous and based on a small sample size. The clearer results of figures 4-6 may provide a more exciting angle where emphasis may want to be put. Below major and minor issues for the authors to consider.

Major issues:

1. In this study there are two regenerative mammals and two non-regenerative mammals. The parameters quantified in Figures 1-2, which are days in culture at which it reached stasis, population doubling time and percentage of proliferative cells, do not neatly divide into the two categories. Rat cells are similar to the spiny mouse and rabbit cells in these parameters. In fact, it doesn't appear to be correlation of regeneration with proliferative capacity in the small sample size that has been tested (out of two non-regenerative mammals, one does not show enhanced proliferative capacity). Thus, while this would have been a significant result given the earlier proposed models of regeneration, it appears that the proposed claim "Enhanced proliferative ability of fibroblasts is associated with, but not restricted to, regenerating mammals." cannot be made based on the provided evidence. Similarly, in Figure 3, enhanced proliferative capacity also does not segregate between regenerative and non-regenerative mammals and the Mus cells show higher senescence markers sooner than the other three species. Thus, currently these results need to be further strengthened with larger sample size as it currently does not show significant correlation (or anti) of proliferative capacity and regenerative potential.
2. It is possible that the problem of comparing rat and rabbit (the two highly proliferative cells by figure 2) with mouse and spiny mouse (the lower proliferative cells) could be addressed by looking at different ages of the animal from which the cells were sourced. From the methods section, the rat and rabbit cells came from much older animals, and perhaps this could be having an effect. However, as stated, the conclusions currently drawn do not seem to match the data.
3. Given that the response to oxidative stress is the strongest mechanistic function that is significantly different between regenerating and non-regenerating mammals in the paper, a few more experiments and explanations are important. Eg: Are there structural features (bioinformatically) unique to Glutathione Peroxidase in regenerating mammals that explain its higher activity under oxidative stress when compared to closely-related non-regenerating species?
4. Are mitochondrial structure and potential also protected in regenerative mammals under oxidative stress: what causes impaired mitochondrial activity under oxidative stress and how is it prevented in regenerative mammals?

Minor issues:

- It may not be clear to all readers that rabbit ear is a model of scar-less regeneration and so the authors may profit from stating this more clearly in the abstract/introduction.
- Many sentences could be difficult to follow due to phrases that began with contrasting the previous sentence but that were in fact agreeing on and building on it. E.g. "In contrast, exposure to Mus cells to sub-lethal concentrations of H₂O₂...".

Reviewer #3:

Remarks to the Author:

Summary:

The authors use a comparative approach between *Acomys* (regenerator) and *Mus* (non-regenerator), later expanded to include rabbit (regenerator) and rat (non-regenerator) to characterize some aspects of the response to injury across species. Their main conclusions are a) *Acomys* fibroblasts are more resistant to senescence than *Mus* fibroblasts, as indicated by SD-Bgal, H2AX, p19 and p16 and downstream markers of stress induced senescence such as p21 and p53; b) exposure to H₂O₂ induces p53, p21, p16, p19, and senescence in mouse and rat cells, whereas ear fibroblasts from spiny mice and rabbits do not activate these tumor suppressors and

are refractory to cellular senescence; c) regenerator species (in contrast to non-regenerator species) react to gamma irradiation by upregulating p53 and p21 but not p16 and p19; d) intracellular hydrogen peroxide overwhelms compensatory mechanisms that protect regenerating species, and that subsequent mitochondrial disruption triggers cellular senescence in non regenerators (but not regenerators). They conclude that intrinsic cellular differences between species are partially responsible for their different responses to wounding and that injury- induced cell proliferation is partly controlled by differential regulation of key tumor suppressor proteins and differential responses to ROS induced senescence.

Review:

This manuscript centers on an important question in the field of mammalian regeneration: what are the mechanisms that underlie variability in the mammalian response to injury, with most species responding to injury with fibrotic scarring and some with a regenerative response. The main question is to what extent these different responses are due to species-specific intrinsic cell differences and what are the main mechanisms involved in early response to injury.

The use of a comparative approach is correct and indeed a strength of the system established by the Seifert group. This is an interesting, well written paper, on an important subject, supported by good quality, quantitative data subject to strong statistical treatment and a solid discussion which integrates the observations within the conceptual framework of the field. Given that the two mammalian regenerative species (Acomys and rabbit) are not well developed, genetically tractable models, in vivo work is mostly limited to descriptive histology and omics approaches. Therefore, the authors extract cells from tissues, establish in vitro cultures and study the behavior of the isolated populations in terms of senescence, population doublings, proliferation rate, expression of key tumor suppressors and response to stressors (oxidative and gamma irradiation). It concludes that there are indeed some cell intrinsic differences between regenerators and non-regenerators and points to the importance of tumor suppressors and the response to oxidative stress as mechanisms underlying the response. The manuscript is a valuable contribution which extends previous work by the Seifert group in the development of Acomys as a mammalian regeneration model and the comparative approach to mammalian regeneration. It is good quality science that is worthy of publication, although the manuscript can (and should) be improved in several ways.

Concerns:

1) While the paper is well written in terms of structure and syntax, a number of weak points are not addressed or glossed over. The repeated use of the categories 'regenerating mammals' vs 'non regenerating mammals' tends to gloss over the fact that they are not studying 2 categories, but rather exceptions to the norm.

2) The authors extract cells from tissues, establish in vitro cultures and study the behavior of the isolated populations. They implicitly assume that the cell populations being cultured and characterized in vitro are representative of the endogenous population that react to injury in vivo; also, they assume that the cells isolated from the different species are equivalent in identity and function. Their conclusions may not be relevant to the in vivo situation. While the authors obliquely acknowledge this in the last sentence of their discussion, I believe they should explicitly acknowledge that their system is highly artificial (artificiality which is underscored by the fact that the establishment of their in vitro population subjects the cells to massive upheaval via mechanical disruption, chemical digestion and adaptation to 2D culture conditions).

3) SA-Bgal staining is canonically accepted as a marker for senescence in the fields of cancer and aging research, and is used by the authors. However, the SA-Bgal data presented in the paper show some inconsistencies. In Figure 1C, the staining in Mus is what would be expected in senescing cells, with high frequency (80%) and strong signal for SA-Bgal throughout the entire cytoplasm of cells with 'ghost-like' morphology. However, in Acomys, the staining is less frequent (20%) and much weaker. The authors state that 'In contrast to Mus, we observed significantly

fewer SA-βgal+ cells in Acomys cultures regardless of passage number...' and comment that 'Interestingly, Acomys cells that were SA-βgal+ still retained a spindle shaped morphology typical of fibroblasts', which suggests that this is true at all passage numbers, presumable even late ones. Further, in the second paragraph of the section entitled Enhanced proliferative ability of fibroblasts is associated with, but not restricted to, regenerating mammals, the authors state that '...at P9 (mean PD = 17) SA-βgal+ cells accounted for approximately 70% of Acomys cultures...'. These observations throw some doubt as to the significance of SA-βgal staining in Acomys cells. While these doubts are weakened by the subsequent use of other senescence indicators, perhaps the authors address and clarify these points?

3) The authors extend their senescence phenotype characterization using a marker panel consisting of H2AX (marker of DS breaks), p19 & p16 (markers of CDKN2A activation) and p21 & p53 (marker of stress induced senescence). The results, summarized in Figure 11, show high % of positive cells in Mus (but not Acomys) for all markers. This result could indicate high resistance to senescence, as the authors suggest, provided that the antibodies (mostly monoclonal) recognize Acomys epitopes efficiently as they do Mus epitopes, and do not show cross-reactivity. Importantly, this issue affects every Figure in the paper containing IF data. Can the authors comment on how they have determined whether the antibodies used are equally efficient for IF in the different species and recognize the intended targets, without significant cross-reactivity? Have the authors characterized the antibodies by western blot and found clean bands of the appropriate molecular weight in all species (particularly Acomys)?

4) The authors set up a comparison of (Mus and rat) vs (Acomys and rabbit) as examples of non-regenerator vs regenerator categories. Any cellular phenotype partitioning neatly between the two groups can be said to correlate with the category.

The categories partition well between regenerators and non-regenerators in their response to stress by H2O2 and gamma radiation. The same is true regarding analysis of ROS response and mitochondrial activity.

Senescence (assuming antibodies are working correctly) does not partition neatly. At atmospheric O₂, Acomys and Mus score higher than rat and rabbit. This 'partition' is maintained at physiological O₂, the effect of which seems to be to make fibroblasts of all 4 species less SA-βgal + (ie, more resistant to senescence) (compare Fig 2D to Fig 3g).

Proliferation at atmospheric O₂ does not partition neatly. Regarding population doublings, Mus and Acomys score low, while rat and rabbit score high (Fig 2A). Regarding proliferation rate at P2, Mus is about 25%, but >80% in the other three species. When testing proliferation at physiological O₂, Mus and Acomys show increased population doublings while rat and rabbit do not. And Mus increases its proliferation rate from 20% at atmospheric O₂ to 75% at physiological O₂, while the other 3 species are relatively unaffected (Fig 3F).

It must be asked: while it sounds reasonable that a regenerating species must have good proliferation capacity and resist senescence in response to wounding, how can it be explained that while rat proliferates and resists senescence better than Acomys regardless of O₂ tension, it is Acomys that regenerates while rat does not? It would seem this fact suggests an important part of the puzzle is missing. This should be specifically addressed in the discussion.

5) The title of the section 'Physiological oxygen (3%) significantly enhances proliferative capacity of Acomys fibroblasts' is a bit misleading, as it is quickly stated that it has the same effect on Mus fibroblasts. Please consider revising.

6) Furthermore, in this same section, the authors state that 'During regeneration or fibrotic repair of external tissues, resident cells will experience oxygen levels closer to physiological O₂ after a

brief exposure to atmospheric oxygen levels.' This could be considered something of an oversimplification. The initial O₂ tension in the wound could be a complex response depending happens inside the tissue could depend strongly on the response of the tissue... for example, increased consumption of metabolic O₂ or loss of microcirculation due to clotting could create a hypoxic event... or, atmospheric O₂ could diffuse inward from the cut creating a gradient, with microgradients depending on vessel distribution and blood flow... Please consider revising.

6) I would suggest a histogram showing quantitative results be added to Figure 6A.

7) Result section, line 2: the authors state that 'connective tissue fibroblasts are the dominant source for blastema cells' to justify their choice of focusing on fibroblasts in their study. They provide a reference (21, Muneoka et al) from a study in axolotl limb regeneration from 1986. I would suggest reinforcing this with more recent and relevant references.

8) The characterization of fibroblasts is limited to IF with vimentin. Given that the whole paper is based on a comparison of cell populations between species, the paper could be strengthened by expanding the characterization with other fibroblast markers.

9) An evaluation of telomere length across species and time points would further inform the conclusions on senescence.

Reviewer: Gustavo Tiscornia

Reviewer #4:

Remarks to the Author:

General comment.

In this manuscript Saxena et al. address the role of fibroblast intrinsic properties in the regenerative capacity of mammals. To this end, they compared cultured fibroblasts from regenerating (African spiny mouse, rabbit) and non-regenerating (mouse, rat) mammal species for their senescence phenotype and resistance to oxidative stress, in search for possible correlations. The hypothesis that the autonomous activity of fibroblasts is central to skin regeneration, and more generally to epimorphic regeneration of any organ or appendage, is interesting. However, the manuscript does not bring clear answers to the questions raised by this interesting hypothesis.

The idea that differences in proliferative ability or inclination towards senescence may distinguish regenerative from non-regenerative mammals was attractive, but proved largely wrong, as acknowledged by the authors, and shown at length in fig. 1, 2 and 3. This could have been a good reason to try an unbiased approach, comparing the fibroblasts of the 4 species, in real conditions of regeneration, via RNA sequencing and/or proteomic analysis. Rather, the authors looked for another candidate, and they chose the resistance to oxidative stress. They did find some correlation here, but the relevance of their observations to the physiological process of regeneration is quite disputable (see "major specific comments"). In addition, conclusions are flawed by inappropriate experimental conditions (see "major specific comments"). In addition, the possible link between oxidative stress-induced senescence and regeneration has not been tested by in vivo experiments in regenerative and non-regenerative species. I do not believe that, in its present form, the manuscript meets the standard of Nature Communication.

Major specific comments

- Throughout the manuscript, fibroblast populations are treated as if they were homogenous. Dermis fibroblasts (for instance) are indeed heterogenous, and this is particularly interesting in the context of regeneration, though not taken into consideration here, which is unfortunate.

- in the paragraphs comparing only *Mus* and *Acomys* ("Sub-lethal H₂O₂ exposure..." pp.10-12) or all 4 species ("Fibroblasts from regenerating ..." pp.12-13) and the associated fig.4 and 5, significant correlation is found between the origin of fibroblasts (regenerative or non-regenerative species) and oxidative stress resistance. Unfortunately, this correlation is observed after choosing the maximum dose of H₂O₂ (300 μ M), leading to what is now appropriately understood as "oxidative distress" (Sies H., *Redox Biol.* 2017, 11, 613–619), which is confirmed by the analysis of DNA damage. Yet, recent publications point instead to oxidative eustress (or redox signaling) as being relevant for the induction of regeneration.

- Moreover (same paragraphs and fig.4-5), there is no attempt to quantify the amount of H₂O₂ having entered fibroblasts after external treatment. This is an important point since the observed resistance could be due to defective peroxide import. Absence of DNA damage (for instance) would necessarily result from such an impairment.

- The same weakness affects the results in fig.6, when the authors want to "track the strength and duration of intracellular H₂O₂ levels". Here for the first time, the authors attempt to have a rough idea of H₂O₂ intracellular levels after treatment, by imaging PO1 fluorescence. This type of imaging is not easily quantitative, and the manuscript does not even try. There are much better and well-established techniques to quantify intracellular H₂O₂ levels, using genetically-encoded fluorescent sensors (as developed by the groups of V. Belousov or T.P. Dick for instance), readily amenable to fibroblasts in culture.

- The situation is compounded when it comes to the intracellular distribution and the dynamics of H₂O₂ levels (panel A in fig.6A). The poor images presented only suggest a defect in H₂O₂ import in the fibroblasts from regenerating species, but even this should be rigorously tested. In addition, the authors should at least comment about their hypothesis, if any, about the source of H₂O₂ present 48h after treatment.

Other specific comments

- The discussion should be careful enough to distinguish regeneration process between adult and larvae, because the spatial distribution of H₂O₂ and its requirement are clearly different between the two stages. Important literature concerning H₂O₂ involvement in regeneration in vertebrates is also missing (Han et al. *Cell Res* 2014; Chen et al. *Dev Cell* 2016; Labit et al. *Sci Rep* 2018; Ferreira et al. *Nat Com* 2018 ...)

- γ H2AX is not a specific marker of senescent cells (even including an EdU test as performed), as said by the authors, but marks any Double Strand Break, including the ones which will be eventually repaired in recovering cells, and in cells which will eventually die rather than enter senescence. P21 is not a reliable marker of senescence either (review by Sharpless and Sherr, 2015, *Nature Rev Cancer*).

- Catalase activity measurement is highly indirect and reflects H₂O₂ consumption in total cellular extracts, where catalase is not the sole actor. It is also dubious whether total catalase content (essentially within lysosomes) is relevant here.

- Members of the Gpx family are indeed "antioxidant scavengers", but not only. GPx activity is highly dependent on the overall redox potential, and also participate to redox signaling. In addition, H₂O₂ is the preferred (but not unique) substrate for GPx1, but not for other GPx measured in the same assay.

- Typing errors should be corrected (e.g. in the abstract, doubling of "in" l.2-3, one word missing in the sentence l.10-13 ...)

- ref.44 is identical to ref.5.

Consensus Responses to Reviewers

We appreciate the comments from our reviewers. We were delighted that all four reviewers found our manuscript interesting and the data of high quality. In kind, we have done our best to address each comment individually. Where appropriate we indicate where we have changed the manuscript and added new data and analyses.

Reviewer #1

This ms. by Seifert and colleagues investigates the basis for differential regenerative capacity between different mammals (using cultured cells as a model system), focusing on differences (and underlying basis) between propensity to undergo senescence.

Comparing senescence, proliferative capacity between different regenerative and non-regenerative they find inhibition of senescence in cells from regenerative mammals, but proliferative capacity does not correlate with regenerative capacity. Further investigating the mechanistic basis for regenerative differences the authors describe increased stress resistance to oxidative stress (H2O2) induced senescence, accompanied by increased levels of catalase and an insensitivity of mitochondria to oxidative stress.

This study is undoubtedly interesting with potentially important implications. However, the study suffers from being wholly descriptive - its unclear whether many of the effects shown here (e.g. catalase upregulation/oxidative stress/response of mitochondria to oxidative stress) are relevant for the differences in regenerative capacity since no experiments have been done to directly address this. Key points the authors may want to consider are the following.

We thank the reviewer for pointing the important implications of our work.

R1.1 The authors show nicely that cells from regenerative animals are resistant to oxidative stress induced senescence (through addition of H2O2) - how relevant this is for the regenerative difference between cells/organism is unclear. Is there increased oxidative stress in the cell lines reported, and (more importantly) does manipulation of endogenous oxidative stress levels affect regenerative capacity? e.g. does ROS scavenging significantly impede senescence in cells for non-regenerative animals

The cells used in this study were primary fibroblasts from the ear pinna. Each biological replicate represents an independent line from a single animal. In our initial submission we included data using the chemical H2O2 sensor PO1 on untreated, growing cells. This data, which is now included in Supplementary Figure 2, suggested that baseline levels of oxidative stress are similar across species, at least as it pertains to H₂O₂ levels. In addition to this data, we now include a genetic hydrogen peroxide sensor (HyPer, Belousov et al., 2006, *Nature Methods*). Similar to the PO1 data, on average, cells at rest show similar levels of H₂O₂ across species.

With regard to the second part of the question, we have conducted a series of experiments using N-acetylcysteine (NAC). NAC is the N-acetyl derivative of cysteine and as a glutathione precursor, when added to cells it will increase glutathione (GSH) stores. Increasing GSH levels effectively increases glutathione peroxidase (Gpx) activity which reduces H₂O₂ to produce H₂O and an oxidized GSH (for example see Brigelius-Flohe, 1999, Elbini et al., 2016 and other references). Thus, we pre-treated cells with NAC and then administered H₂O₂ to test if increasing ROS scavenging can impede senescence. Indeed, this new data shows that increased

glutathione protects cells from the negative effects of exogenous ROS and prevents ROS-induced senescence as measured by p21 and SA- β gal. We now include this data in a new Figure 6 and Supplementary Figure 4)? We appreciate the request for additional *in vivo* experiments to examine how ROS scavenging may directly affect regenerative capacity. We intend to conduct these experiments as part of a larger set of *in vivo* studies that will form the basis for another manuscript. Thus, we believe these experiments are beyond the scope of the current manuscript which instead, leverages *in vitro* and *in vivo* data to provide the mechanistic basis for conducting those studies.

R1.2 *In the same line of determining cause and effect, Figure 6 title states "Increased GPx activity and mitochondrial resilience protect regenerating species from stress-induced senescence." - this is not what is shown in figure 6, merely that there is a correlation.*

First, we thank the reviewer for calling to our attention the imprecise wording used in the previous title of that figure. We agree that alone, our measurements of mitochondrial resilience and of Gpx activity do not directly tie the cytoprotection we observe in fibroblasts from regenerating species to the increased stress resistance phenotype. Similar to our response above, to address this shortcoming and directly tie our Gpx finding to mitochondrial stress resistance we performed a series of experiments using a genetic H₂O₂ sensor (HyPer and N-acetylcysteine (NAC)). We include all of this data in a revised Figure 6. First, we now present data for cells from all four species using HyPer prior to and following exposure to H₂O₂. This demonstrates that fibroblasts from regenerating species have a stronger scavenging system that more quickly detoxifies excess H₂O₂. We then present our original data showing that GPx activity (two scavenging systems which reduce H₂O₂) is significantly increased in cells from regenerating species. In an attempt to more directly tie GPx activity to stress-induced senescence we treated *Mus* and *Rattus* cells with H₂O₂ in the presence or absence of NAC and measured oxygen consumption, maximal respiration, ATP production and spare respiratory capacity. We also measured induction of senescence via p21 and SA- β gal. These data revealed that increasing GPx activity via NAC could protect cells (and their mitochondria) from ROS stress and that cells were protected from senescence similar to what we observed in cells from regenerating species.

Reviewer #2

In this manuscript, Saxena et al. address some of the mechanistic differences between fibroblasts from regenerative versus related non-regenerative mammals. As the authors identify in references (e.g. Bryant et al. 1986), fibroblasts likely play an important role in the regenerative capacity of a tissue and therefore mechanistic insight on their cellular differences, correlated with that capacity, is an important area of research.

By using freshly isolated fibroblast cells, the authors aimed to exclude the variables in regenerative capacity due to tissue environment in the different mammals and focus on the intrinsic properties of the fibroblasts. To this end, the authors began by measuring the senescence and proliferation rates of cultured cells as a function of time in the typical models of the lab mouse and rat, and also the regenerative models of the African spiny mouse and the rabbit ear. The authors used culture conditions (3% oxygen) that improved the proliferative capacity of the mouse and rat cells, and which led them to investigate the effect of increased oxygen stress (H₂O₂), as well as gamma radiation, on the fibroblasts. Finally, the authors identify a mechanistic difference between the regenerative and non-regenerative cells

(glutathione peroxidase), which can account for the observed different responses to the oxidative stress.

R2.1. *The authors show some very interesting findings on the increased resistance to oxidative damage and gamma radiation damage in the fibroblasts which segregate neatly between regenerative and non-regenerative species. However, the main conclusion that the authors draw from Figures 1 to 3 about proliferative capacity and regenerative potential seem ambiguous and based on a small sample size. The clearer results of figures 4-6 may provide a more exciting angle where emphasis may want to be put.*

We included the data in Figures 1-3 because we believe it is important to make the point that the ear pinna fibroblasts do not neatly segregate across healing phenotype with respect to proliferative capacity and reluctance to senesce. However, we agreed with this reviewer that the data and results section for Figures 1-2 could be pared down and thus we have shortened this part of the manuscript and moved some of this data into the Supplementary material. Our new Figures 1 & 2 show proliferative ability and senescence across all four species. As you will see, we have also expanded the second section of the manuscript that focused on stress resistance.

R2.2. *In this study there are two regenerative mammals and two non-regenerative mammals. The parameters quantified in Figures 1-2, which are days in culture at which it reached stasis, population doubling time and percentage of proliferative cells, do not neatly divide into the two categories. Rat cells are similar to the spiny mouse and rabbit cells in these parameters. In fact, it doesn't appear to be correlation of regeneration with proliferative capacity in the small sample size that has been tested (out of two non-regenerative mammals, one does not show enhanced proliferative capacity). Thus, while this would have been a significant result given the earlier proposed models of regeneration, it appears that the proposed claim "Enhanced proliferative ability of fibroblasts is associated with, but not restricted to, regenerating mammals." cannot be made based on the provided evidence. Similarly, in Figure 3, enhanced proliferative capacity also does not segregate between regenerative and non-regenerative mammals and the Mus cells show higher senescence markers sooner than the other three species. Thus, currently these results need to be further strengthened with larger sample size as it currently does not show significant correlation (or anti) of proliferative capacity and regenerative potential.*

We appreciate the concern raised by the reviewer. However, as stated above, we believe that it is important to include data that tested hypotheses proposed by other researchers (e.g., Pajcini et al., 2010, Hesse et al., 2015, Tanaka et al., 1997) where the idea was raised that proliferative capacity of fibroblasts is tied to regenerative ability. In fact, our data shows that this is only partly true. In our study, cells from both regenerating species show enhanced proliferative capacity compared to *Mus* (and to human) with respect to time to stasis and population doublings (new Figure 1). But as pointed out by the reviewer, cells from *Rattus* break this association. Thus, we stated that "*Enhanced proliferative ability of fibroblasts is **associated with**, but not restricted to, regenerating mammals*". This is a true statement based on our four species comparison and data from the literature. With respect to sample sizes, we are constrained by the number of bonafide mammalian regeneration models which currently stands at two. We appreciate that we must be careful in drawing broader inferences from our data. As we replied in comment R2.1, we have reduced this result section and have made an effort to more clearly represent these findings. For example, we have re-titled the revised first section of the paper, "*In*

vitro proliferative ability of fibroblasts is not restricted to regenerating mammals”. While enhanced proliferative capacity does not correlate perfectly with regenerative ability (or lack thereof), cells from both regenerating species do show that they proliferate well in culture and escape stasis.

R2.3. *It is possible that the problem of comparing rat and rabbit (the two highly proliferative cells by figure 2) with mouse and spiny mouse (the lower proliferative cells) could be addressed by looking at different ages of the animal from which the cells were sourced. From the methods section, the rat and rabbit cells came from much older animals, and perhaps this could be having an effect. However, as stated, the conclusions currently drawn do not seem to match the data.*

Our cross-species comparison reflects cells exhibiting species-specific growth parameters. As stated in our methods, mouse cells were the youngest and as such, would be expected to have the fastest doubling rates in culture (Parrinello et al., 2003, *Nature Cell Biology*). Instead they have the slowest. Furthermore, proliferative ability of mammalian fibroblasts in culture is independent of lifespan (Patrick et al., 2016. *Ageing*). For these reasons we do not believe animal age is driving the observed differences in proliferative rate. While we could collect additional aging data, we believe it would not fit into the scope of this manuscript. Similar to the previous comment, we have altered our phrasing in stating our conclusion.

R2.4 *Given that the response to oxidative stress is the strongest mechanistic function that is significantly different between regenerating and non-regenerating mammals in the paper, a few more experiments and explanations are important. Eg: Are there structural features (bioinformatically) unique to Glutathione Peroxidase in regenerating mammals that explain its higher activity under oxidative stress when compared to closely-related non-regenerating species?*

We appreciate this question from the reviewer. At the reviewer’s suggestion, we have directly compared the sequences of GPx1, GPx2 and GPx3 using the Multiple Sequence Alignment tool in CLUSTALW and present this data as a new Supplementary Figure 3. This data reveals the highly conserved nature of glutathione peroxidase across species: percentage similarity >82% for GPx1 and >88% for GPx2 and GPx3. While it is possible that single sequence differences could potentially account for small differences in enzyme activity, it is much more likely that increased GPx activity observed in response to H₂O₂ exposure in the regenerating species occurs via faster replenishment of depleted GSH stores (or simply greater GSH recycling). In fact, our NAC experiments support this explanation.

R2.5 *Are mitochondrial structure and potential also protected in regenerative mammals under oxidative stress: what causes impaired mitochondrial activity under oxidative stress and how is it prevented in regenerative mammals?*

This is an important point and we have directly addressed this through a new series of experiments. In addition to quantifying basal respiration, ATP production, maximal respiration and spare respiratory capacity from intact cells treated with H₂O₂ (compared to control cells), we performed a similar experiment, but instead isolated mitochondria as outlined in our methods section (see *Mitochondrial isolation*). After isolating mitochondria, we assessed mitochondrial

function using the Seahorse analyzer and exposed mitochondria to Pyruvate/Malate and Adenosine diphosphate (ADP) via port A to measure ATP production (State III); Oligomycin to measure proton leak across inner mitochondrial membrane (State IV)); FCCP via port C to induce State V_{CI} and (Rotenone + Succinate) to induce State V_{CII} respiration (measures maximum electron transport system capacity thru complex I and complex II respectively). While we found a significant decrease in state III respiration for mitochondria from *Mus* and *Rattus*, we observed no change in state III respiration for *Acomys* and *Oryctolagus*. Analyzing the RCR, we found that it too was significantly decreased in mitochondria from *Mus* and *Rattus* after H_2O_2 treatment. Again, and in contrast to the non-regenerating mammals, we found no significant change in the RCR for mitochondria from *Acomys* and *Oryctolagus*. We now present these results in a new Figure 5 which show that mitochondria from regenerating species exhibit the same protection in structure and function that we observed for intact cells. This is highlighted by the lack of change in coupling (measured as RCR) that demonstrates the mitochondrial membrane is structural intact (State IV component of RCR) and membrane potential is maintained as measured by State III response.

With respect to second question, while we agree that this is an important question, we also believe additional experiments along these lines are beyond the scope of the present study. However, the new data we provide in Figure 6 draws a link to increased glutathione peroxidase activity which we show is significantly enhanced in regenerating species compared to non-regenerating species. Furthermore, increasing GPx activity via NAC pretreatment protects cells in non-regenerating species from stress-induced senescence and mitochondrial dysfunction.

Minor issues:

R2.6. *It may not be clear to all readers that rabbit ear is a model of scar-less regeneration and so the authors may profit from stating this more clearly in the abstract/introduction.*

This is good point. We have now made this clearer in the second to last paragraph of the introduction.

R2.7. *Many sentences could be difficult to follow due to phrases that began with contrasting the previous sentence but that were in fact agreeing on and building on it. E.g. "In contrast, exposure to *Mus* cells to sub-lethal concentrations of H_2O_2 ..."*

We appreciate the reviewer for pointing this out. We have combed through the manuscript and made sure to change the language to avoid these contradictory sentences.

Reviewer #3.

Summary:

*The authors use a comparative approach between *Acomys* (regenerator) and *Mus* (non-regenerator), later expanded to include rabbit (regenerator) and rat (non-regenerator) to characterize some aspects of the response to injury across species. Their main conclusions are a) *Acomys* fibroblasts are more resistant to senescence than *Mus* fibroblasts, as indicated by *SD-Bgal*, *H2AX*, *p19* and *p16* and downstream markers of stress induced senescence such as *p21* and *p53*; b) exposure to H_2O_2 induces *p53*, *p21*, *p16*, *p19*, and senescence in mouse and rat cells, whereas ear fibroblasts from spiny mice and rabbits do not activate these tumor suppressors and are refractory to cellular senescence; c) regenerator species (in contrast to non-regenerator species) react to gamma irradiation by upregulating *p53* and *p21* but not *p16* and *p19*; d) intracellular hydrogen peroxide overwhelms compensatory mechanisms that protect regenerating species, and that subsequent mitochondrial*

disruption triggers cellular senescence in non regenerators (but not regenerators). They conclude that intrinsic cellular differences between species are partially responsible for their different responses to wounding and that injury- induced cell proliferation is partly controlled by differential regulation of key tumor suppressor proteins and differential responses to ROS induced senescence.

Review

This manuscript centers on an important question in the field of mammalian regeneration: what are the mechanisms that underlie variability in the mammalian response to injury, with most species responding to injury with fibrotic scarring and some with a regenerative response. The main question is to what extent these different responses are due to species-specific intrinsic cell differences and what are the main mechanisms involved in early response to injury.

The use of a comparative approach is correct and indeed a strength of the system established by the Seifert group. This is an interesting, well written paper, on an important subject, supported by good quality, quantitative data subject to strong statistical treatment and a solid discussion which integrates the observations within the conceptual framework of the field. Given that the two mammalian regenerative species (Acomys and rabbit) are not well developed, genetically tractable models, in vivo work is mostly limited to descriptive histology and omics approaches. Therefore, the authors extract cells from tissues, establish in vitro cultures and study the behavior of the isolated populations in terms of senescence, population doublings, proliferation rate, expression of key tumor suppressors and response to stressors (oxidative and gamma irradiation). It concludes that there are indeed some cell intrinsic differences between regenerators and non-regenerators and points to the importance of tumor suppressors and the response to oxidative stress as mechanisms underlying the response. The manuscript is a valuable contribution which extends previous work by the Seifert group in the development of Acomys as a mammalian regeneration model and the comparative approach to mammalian regeneration. It is good quality science that is worthy of publication, although the manuscript can (and should) be improved in several ways.

Concerns

R3.1. While the paper is well written in terms of structure and syntax, a number of weak points are not addressed or glossed over. The repeated use of the categories ‘regenerating mammals’ vs ‘non regenerating mammals’ tends to gloss over the fact that they are not studying 2 categories, but rather exceptions to the norm.

We agree with the reviewer that the two regenerating species may be exceptions to the norm. We have tried to rely on the species names when drawing contrasts, but we still do refer to “regenerating” and “non-regenerating” mammals in the text. This is the least cumbersome use of language for the purposes of this manuscript.

R3.2. The authors extract cells from tissues, establish in vitro cultures and study the behavior of the isolated populations. They implicitly assume that the cell populations being cultured and characterized in vitro are representative of the endogenous population that react to injury in vivo; also, they assume that the cells isolated from the different species are equivalent in identity and function. Their conclusions may not be relevant to the in vivo situation. While the authors obliquely acknowledge this in the last sentence of their discussion, I believe they should explicitly acknowledge that their system is highly artificial (artificiality which is underscored by the fact that the establishment of their in vitro population subjects the cells to massive upheaval via mechanical disruption, chemical digestion and adaptation to 2D culture conditions.

We appreciate these points which we tried to address in the manuscript. The primary ear pinna cells we are studying are the resident connective tissue cells of the ear and are the most populous cells that respond to injury *in vivo*. We acknowledge they are a heterogeneous population of cells in the text, but also use vimentin to show that >95% are fibroblasts. Although we assumed the reader would implicitly understand the caveats of culturing cells vs. analyzing them *in vivo*, the reviewer raises a fair point and we now include two additional statements: one in the introduction indicating that we chose to study cells *in vitro* so we could explicitly look at intrinsic cellular features under controlled conditions and a second sentence in the last paragraph pointing out the caveat of extrapolating *in vitro* to *in vivo*.

R3.3 SA-Bgal staining is canonically accepted as a marker for senescence in the fields of cancer and aging research, and is used by the authors. However, the SA-Bgal data presented in the paper show some inconsistencies. In Figure 1C, the staining in Mus is what would be expected in senescing cells, with high frequency (80%) and strong signal for SA-Bgal throughout the entire cytoplasm of cells with 'ghost-like' morphology. However, in Acomys, the staining is less frequent (20%) and much weaker. The authors state that 'In contrast to Mus, we observed significantly fewer SA-βgal+ cells in Acomys cultures regardless of passage number...' and comment that 'Interestingly, Acomys cells that were SA-βgal+ still retained a spindle shaped morphology typical of fibroblasts', which suggests that this is true at all passage numbers, presumable even late ones. Further, in the second paragraph of the section entitled Enhanced proliferative ability of fibroblasts is associated with, but not restricted to, regenerating mammals, the authors state that '...at P9 (mean PD = 17) SA-βgal+ cells accounted for approximately 70% of Acomys cultures...'. These observations throw some doubt as to the significance of SA-Bgal staining in Acomys cells. While these doubts are weakened by the subsequent use of other senescence indicators, perhaps the authors address and clarify these points?

We thank the reviewer for highlighting this point. Our protocol for staining cells with SA-βgal is robust and worked well across species. Our revised manuscript now only includes cells cultured at physiological oxygen (except for one graph showing population doublings). Some of what the reviewer is referring to were early images acquired under 20%O₂. We have tried to provide better images of the SA-βgal staining in Figure 2. We believe these images are more representative of the staining we see throughout all our experiments. With respect to the P9 statement, this refers to cells being cultured in ambient oxygen. This result reflects that *Acomys* cells are more sensitive to oxygen concentration than either *Rattus* and *Oryctolagus* and thus at P9 (close to when they reach stasis) 70% of the cells stain positive for SA-βgal. We believe this shows that the staining works well and mirrors what happens as cells enter senescence.

R3.4 The authors extend their senescence phenotype characterization using a marker panel consisting of γ-H2AX (marker of DS breaks), p19 & p16 (markers of CDKN2A activation) and p21 & p53 (marker of stress induced senescence). The results, summarized in Figure II, show high % of positive cells in Mus (but not Acomys) for all markers. This result could indicate high resistance to senescence, as the authors suggest, provided that the antibodies (mostly monoclonal) recognize Acomys epitopes efficiently as they do Mus epitopes, and do not show cross-reactivity. Importantly, this issue affects every Figure in the paper containing IF data. Can the authors comment on how they have determined whether the antibodies used are equally efficient for IF in the different species and recognize the intended targets, without significant

*cross-reactivity? Have the authors characterized the antibodies by western blot and found clean bands of the appropriate molecular weight in all species (particularly *Acomys*)?*

We agree that it is important to validate antibodies for cross-species experiments and we have done our best to do so. To address the reviewer's comment, we now include a series of sequence alignments between the four species for each protein to which the antibodies are directed. This is presented in Supplementary Figure 6 as two tables and CLUSTAL alignments. The tables show sequence similarity for the entire protein sequence and also for those sequences where specific peptide sequences were used as the antigen. The alignments reveal the very high amino acid sequence similarity across species. Along with these data, we now include additional microscopy images (Figures 2 & 4 and Supplementary Figures 1 and 2) showing cellular localization of the antibodies in positive and negative cells. Our new Supplementary Figure 2 is especially informative as it shows upregulation of p21 and p53 in response to gamma radiation. That SA- β gal and γ -H2AX support our findings using these markers provides further support for their effective use across species. We were not able to validate for all antibodies using western blots for several reasons. First, as our data shows, spiny mice and rabbits normally express very low levels of these markers even in response to H₂O₂ treatment. Thus, acquiring protein lysates with high enough protein concentrations to run on a gel was difficult. Second, several of the polyclonal antibodies were not optimized for western blots and were not reliable even for mouse.

*R3.5. The authors set up a comparison of (*Mus* and *rat*) vs (*Acomys* and *rabbit*) as examples of non-regenerator vs regenerator categories. Any cellular phenotype partitioning neatly between the two groups can be said to correlate with the category. The categories partition well between regenerators and non-regenerators in their response to stress by H₂O₂ and gamma radiation. The same is true regarding analysis of ROS response and mitochondrial activity. Senescence (assuming antibodies are working correctly) does not partition neatly. At atmospheric O₂, *Acomys* and *Mus* score higher than *rat* and *rabbit*. This 'partition' is maintained at physiological O₂, the effect of which seems to be to make fibroblasts of all 4 species less SD-Bgal + (ie, more resistant to senescence) (compare Fig 2D to Fig 3g). Proliferation at atmospheric O₂ does not partition neatly. Regarding population doublings, *Mus* and *Acomys* score low, while *rat* and *rabbit* score high (Fig 2A). Regarding proliferation rate at P2, *Mus* is about 25%, but >80% in the other three species. When testing proliferation at physiological O₂, *Mus* and *Acomys* show increased population doublings while *rat* and *rabbit* do not. And *Mus* increases its proliferation rate from 20% at atmospheric O₂ to 75% at physiological O₂, while the other 3 species are relatively unaffected (Fig 3F). It must be asked: while it sounds reasonable that a regenerating species must have good proliferation capacity and resist senescence in response to wounding, how can it be explained that while *rat* proliferates and resists senescence better than *Acomys* regardless of O₂ tension, it is *Acomys* that regenerates while *rat* does not? It would seem this fact suggests an important part of the puzzle is missing. This should be specifically addressed in the discussion.*

This is the puzzle that our paper lays out in the first part of the results and which we believe the ROS exposure experiments help to explain. Our findings suggest that stress exposure upon injury in the form of H₂O₂ and other ROS induces a senescent (anti-proliferative) phenotype in *Mus* and *Rattus* cells. Our results showing enhanced glutathione peroxidase activity in cells from regenerating species provides evidence of a cytoprotective mechanism and our NAC

experiments in cells from *Rattus* supports that enhanced glutathione peroxidase activity can protect these cells from ROS-induced senescence. We address this specifically in the discussion in light of our new data.

R3.6. *The title of the section ‘Physiological oxygen (3%) significantly enhances proliferative capacity of Acomys fibroblasts’ is a bit misleading, as it is quickly stated that it has the same effect on Mus fibroblasts. Please consider revising.*

We thank the reviewer for this suggestion. We have greatly paired down the first part of our results section and no longer have this title.

R3.7. *Furthermore, in this same section, the authors state that ‘During regeneration or fibrotic repair of external tissues, resident cells will experience oxygen levels closer to physiological O₂ after a brief exposure to atmospheric oxygen levels.’ This could be considered something of an oversimplification. The initial O₂ tension in the wound could be a complex response depending happens inside the tissue could depend strongly on the response of the tissue... for example, increased consumption of metabolic O₂ or loss of microcirculation due to clotting could create a hypoxic event... or, atmospheric O₂ could diffuse inward from the cut creating a gradient, with microgradients depending on vessel distribution and blood flow... Please consider revising.*

Again, much of this section has been reduced and we have revised our statements regarding oxygen concentration to simply state that we chose to use physiological relevant oxygen concentrations because some cell types are sensitive to ambient oxygen concentration.

R3.8 *I would suggest a histogram showing quantitative results be added to Figure 6A.*

We address this comment in response to Reviewer 4 below. Briefly, we now provide a histogram alongside our new data using a genetic oxygen sensor (HyPer).

R3.9. *Result section, line 2: the authors state that ‘connective tissue fibroblasts are the dominant source for blastema cells’ to justify their choice of focusing on fibroblasts in their study. They provide a reference (21, Muneoka et al) from a study in axolotl limb regeneration from 1986. I would suggest reinforcing this with more recent and relevant references.*

That is correct. Connective tissue cells are overrepresented in blastemas formed from vertebrate appendages because they have the greatest lineage plasticity. We have added “appendage” to this sentence along with two more references (Kragl et al., 2009, Rinkevich et al., 2011). Muneoka is the most appropriate reference and these two papers support the finding. Although there have been two recent scRNAseq papers in salamander limb and another in zebrafish fin, you cannot reliably determine relative cell contributions from single-cell data, so those references are not appropriate.

R3.10. *The characterization of fibroblasts is limited to IF with vimentin. Given that the whole paper is based on a comparison of cell populations between species, the paper could be strengthened by expanding the characterization with other fibroblast markers.*

We appreciate the reviewer's comment. Our aim was to broadly compare heterogeneous populations of connective tissue fibroblasts across species, not to parse subpopulations of these cells. There are few antibody markers that can reliably label fibroblasts alone (despite some claims to the contrary). However, vimentin is one that works *in vitro* and works across species. Being limited to commercially available antibodies this is all we have at the moment. Given the extremely high labeling efficiency (>95%) we believe that vimentin appropriately makes the point that our populations are predominantly fibroblasts and largely exclude keratinocytes and endothelial cells.

R3.11. *An evaluation of telomere length across species and time points would further inform the conclusions on senescence.*

This would be appropriate for human cells and some other species of mammals that exhibit replicative senescence in a telomere dependent manner. However, *Mus* and *Oryctolagus* (Steinhert et al., 2002, Forsyth et al., 2005, Gomes et al., 2011) do not exhibit telomere dependent replicative senescence and *Rattus* have extremely long telomeres. Thus, while we agree that looking at telomere biology in *Acomys* could be interesting, we believe it is beyond the scope of the current paper.

Reviewer #4

General comment.

In this manuscript Saxena et al. address the role of fibroblast intrinsic properties in the regenerative capacity of mammals. To this end, they compared cultured fibroblasts from regenerating (African spiny mouse, rabbit) and non-regenerating (mouse, rat) mammal species for their senescence phenotype and resistance to oxidative stress, in search for possible correlations. The hypothesis that the autonomous activity of fibroblasts is central to skin regeneration, and more generally to epimorphic regeneration of any organ or appendage, is interesting.

However, the manuscript does not bring clear answers to the questions raised by this interesting hypothesis.

The idea that differences in proliferative ability or inclination towards senescence may distinguish regenerative from non-regenerative mammals was attractive, but proved largely wrong, as acknowledged by the authors, and shown at length in fig.1, 2 and 3. This could have been a good reason to try an unbiased approach, comparing the fibroblasts of the 4 species, in real conditions of regeneration, via RNA sequencing and/or proteomic analysis. Rather, the authors looked for another candidate, and they chose the resistance to oxidative stress. They did find some correlation here, but the relevance of their observations to the physiological process of regeneration is quite disputable (see "major specific comments"). In addition, conclusions are flawed by inappropriate experimental conditions (see "major specific comments"). In addition, the possible link between oxidative stress-induced senescence and regeneration has not been tested by in vivo experiments in regenerative and non-regenerative species. I do not believe that, in its present form, the manuscript meets the standard of Nature Communication.

Major specific comments

R4.1. *Throughout the manuscript, fibroblast populations are treated as if they were homogenous. Dermis fibroblasts (for instance) are indeed heterogenous, and this is particularly interesting in the context of regeneration, though not taken into consideration here, which is unfortunate.*

We appreciate the reviewer's comment here. As we replied to reviewer 3 (R3.10), we embrace that we are working with a heterogeneous population of cells. In fact, we tried to make this clear in our results section and were clear to state that we isolated connective tissue cells from the ear pinna. However, in light of this comment, we have refined our wording to indicate that we isolated a heterogeneous population of ear pinna connective tissue fibroblasts. Our use of vimentin staining confirms this. We also agree with the reviewer that considering subpopulations of cells in the context of regeneration is interesting. Alongside single-cell data, we are developing labeling strategies and should be better equipped to address this question in future manuscripts.

R4.2 *In the paragraphs comparing only Mus and Acomys ("Sub-lethal H₂O₂ exposure..." pp.10-12) or all 4 species ("Fibroblasts from regenerating ..." pp.12-13) and the associated fig.4 and 5, significant correlation is found between the origin of fibroblasts (regenerative or non-regenerative species) and oxidative stress resistance. Unfortunately, this correlation is observed after choosing the maximum dose of H₂O₂ (300 μ M), leading to what is now appropriately understood as "oxidative distress" (Sies H., Redox Biol. 2017, 11, 613–619), which is confirmed by the analysis of DNA damage. Yet, recent publications point instead to oxidative eustress (or redox signaling) as being relevant for the induction of regeneration.*

We included data in Figure 4 showing an increase in senescent associated markers as a function of H₂O₂ concentration. We chose the a relatively low (75 μ M) sub-lethal concentration based on the literature and then stepped that up to a very high dose (300 μ M). Both *Mus* and *Rattus* cells respond in a concentration dependent fashion to H₂O₂ and we now include this data in Figure 4. In fact, the reason we only present dose data for 300 μ M H₂O₂ is because we wanted to show that even at the highest doses, fibroblasts from regenerating species appeared highly resistant to exogenous H₂O₂. Thus, the correlation with resistance to ROS-induced senescence holds at both low and high doses.

R4.3-5 *Moreover (same paragraphs and fig.4-5), there is no attempt to quantify the amount of H₂O₂ having entered fibroblasts after external treatment. This is an important point since the observed resistance could be due to defective peroxide import. Absence of DNA damage (for instance) would necessarily result from such an impairment.*

The same weakness affects the results in fig.6, when the authors want to "track the strength and duration of intracellular H₂O₂ levels". Here for the first time, the authors attempt to have a rough idea of H₂O₂ intracellular levels after treatment, by imaging PO1 fluorescence. This type of imaging is not easily quantitative, and the manuscript does not even try. There are much better and well-established techniques to quantify intracellular H₂O₂ levels, using genetically-encoded fluorescent sensors (as developed by the groups of V. Belousov or T.P. Dick for instance), readily amenable to fibroblasts in culture.

The situation is compounded when it comes to the intracellular distribution and the dynamics of H₂O₂ levels (panel A in fig.6A). The poor images presented only suggest a defect in H₂O₂ import in the fibroblasts from regenerating species, but even this should be rigorously tested. In addition, the authors should at least comment about their hypothesis, if any, about the source of H₂O₂ present 48h after treatment.

We appreciate the reviewer's points regarding intracellular H₂O₂ levels which is a key element of our paper. We agree that our qualitative assessment could be improved. In response to the reviewer's request, we took their suggestion and employed a genetically encoded H₂O₂ sensor developed by V. Belousov (i.e., HyPer) to quantify H₂O₂ influx and degradation. We used this sensor to conduct the following experiments which are now included in a new Figure 6 alongside our original PO1 experiments which are presented in Supplementary Figure 3. First, we transfected cells from all four species plated in 24-well glass bottomed plates with a HyPer plasmid (pHyPer-cyto). We then live-imaged these cells using a culture-equipped inverted Olympus IX83 microscope with motorized stage before and after exposure to H₂O₂. Conditions in the culture chamber matched our culture conditions throughout the paper (i.e., physiological oxygen). Prior to addition of HyPer, we selected and imaged different fields of cells (multiple fields per well). Next, 300 μ M H₂O₂ was added to the wells and cells were subsequently imaged every 30 min for 18hrs. We measured the fluorescent intensity of individual cells (≥ 20 cells/cell line) at each time point/species (n=3 cell lines/species). We divided fluorescent intensities at successive time points with the pre-treatment fluorescent intensity of the same cell to calculate a fluorescence ratio (F/F₀) of intensities for each cell. This was then averaged over all cells imaged/cell line and then across cells lines/species and compared across species. We now present time step images of the fluorescence intensity for an individual cell/species in Figure 6a and a quantification of the average ratio across multiple cells for each species (n=3/species) in Figure 6b. This data demonstrates that on average, cells from all species experience the same peak exposure to H₂O₂. However, these traces show that cells from regenerating species detoxify H₂O₂ levels at a significantly faster rate compared to non-regenerating species. This data mimics that more qualitative data presented using the P01 chemical sensor.

R4.6. The discussion should be careful enough to distinguish regeneration process between adult and larvae, because the spatial distribution of H₂O₂ and its requirement are clearly different between the two stages. Important literature concerning H₂O₂ involvement in regeneration in vertebrates is also missing (Han et al. Cell Res 2014; Chen et al. Dev Cell 2016; Labit et al. Sci Rep 2018; Ferreira et al. Nat Com 2018 ...)

We agree with the reviewer and have made sure to clearly indicate the difference in our discussion. We appreciate the additional references, with which we are familiar. We have included most at the suggestion of the reviewer where appropriate.

R4.7. GammaH2AX is not a specific marker of senescent cells (even including an EdU test as performed), as said by the authors, but marks any Double Strand Break, including the ones which will be eventually repaired in recovering cells, and in cells which will eventually die rather than enter senescence. P21 is not a reliable marker of senescence either (review by Sharpless and Sherr, 2015, Nature Rev Cancer).

We appreciate these points. We are familiar with the Sharpless review and with the caveats of using markers to indicate senescence. This is why we used a panel of markers in combination throughout the manuscript. γ -H2AX is useful as stated and many authors agree that p21 is a useful endpoint marker when used in combination with other markers because it is a downstream target that functions to block cell cycle progression.

R4.8. *Catalase activity measurement is highly indirect and reflects H₂O₂ consumption in total cellular extracts, where catalase is not the sole actor. It is also dubious whether total catalase content (essentially within lysosomes) is relevant here.*

We agree that the catalase assay is indirect and reflects H₂O₂ consumption, which admittedly we are interested in looking at in the context of that assay. However, given our focus on glutathione peroxidases, we have de-emphasized the catalase result, although we do still present the data because we believe it is relevant.

R4.9. *Members of the Gpx family are indeed "antioxidant scavengers", but not only. GPx activity is highly dependent on the overall redox potential, and also participate to redox signaling. In addition, H₂O₂ is the preferred (but not unique) substrate for GPx1, but not for other GPx measured in the same assay.*

We appreciate the reviewer's point here. We have altered our wording in the result section to reflect this caveat.

R4.10. Typing errors should be corrected (e.g. in the abstract, doubling of "in" l.2-3, one word missing in the sentence l.10-13 ...)

These have been corrected throughout.

R4.11. ref.44 is identical to ref.5.

These have been corrected throughout.

Reviewers' Comments:

Reviewer #1:

Remarks to the Author:

The authors have addressed my main concerns in a satisfactory manner.

Reviewer #2:

Remarks to the Author:

The authors have done a tremendous job updating the focus of the paper to center more on their very interesting findings that are specific to reactive oxygen species (ROS) and providing additional supporting evidence around ROS. This work importantly contributes to the field and should be published. Below are minor points that the authors can consider as they finalize the manuscript for publication.

The authors have updated the language and conclusions surrounding results sections 1 and 2. About the sentence "These data provide evidence that increased proliferative ability under physiological oxygen concentrations is associated with regenerative ability in some mammals". The word association may appear too strong given that based on the proliferation power characterizing these 4 species a chi-squared test will not be able to provide significant evidence that increased proliferative ability is associated with regenerative ability. Perhaps if the authors wish to support this idea based on existing literature, then this statement could be made in the introduction or discussion sections. Instead commenting the results as two contrasting properties would be appropriate: one phenotype that segregates (ROS response), and another that does not (proliferative capacity). Alternatively, the authors could indicate that proliferative capacity seems to be species specific while response to ROS is regeneration-associated.

With respect to the conclusion "Our cross-species comparison reflects cells exhibiting species-specific growth parameters." The cross-species comparisons reflect multiple variables as indicated by the authors, one of which is the species from which the cells are derived. Another of which is the age of the animals they are derived from. In order to better clarify the author's idea to the readers that the dominating difference is the species, the authors could cite in the manuscript the references they provided in the rebuttal: Parrinello et al., 2003, Nature Cell Biology and Patrick et al., 2016. Aging (Patrick et al 2016 is cited already, but for a different purpose).

While ruling out the possible contribution of structural features of the enzyme in its heightened activity in regenerated animal, the authors have suggested an interesting alternate explanation of how the heightened GPx activity in response to stress is achieved. NAC experiments suggest that an altered kinetics of the GSH pathway or larger stores can also lead to protection against H₂O₂. Adding in the discussion about future directions to figure out the kinetics of GSH metabolism/glutathione stores in regenerating versus non-regenerating mammals may provide further satisfaction to the readers.

The experiments about GPx and NAC provide insight into how the resistance to senescence is achieved. They open the door for future experiments beyond this paper where the authors can test this by directly comparing the glutathione stores/GSH kinetics in regenerating versus non-regenerating mammals. It may be worth to expand on this in the discussion.

The new sets of experiments on the structural and functional integrity of mitochondria bring us closer to understanding the intracellular and metabolic differences between regenerating and non-regenerating mammals. Congratulations to the authors on the efforts and the results that more clearly the mechanistic differences between the two groups.

Reviewer #3:

Remarks to the Author:

Saxena et al. present a revised manuscript focusing on characterization of intrinsic differences between fibroblasts obtained from two species capable of regeneration (*Acomys* and rabbit) in comparison to two species known to heal by fibrotic scarring (rat and mouse). They characterize fibroblasts in terms of their proliferative capacity in relation to oxygen tension and resistance to senescence. They extend this comparison to situations of oxidative and gamma radiation induced stress. They complete their characterization by comparing mitochondrial associated phenotypes and mechanisms of detoxification of free radicals, identifying resistance to senescence as a key characteristic of fibroblasts in regenerating mammals. This revised manuscript is significantly strengthened in comparison to their initial submission. They have changed the structure of the text, presented additional data and for the most part, have addressed my concerns to their first manuscript. Their work is very solid, quantitative and statistically sound. Their discussion is well written and integrates their results well into the literature. I find their conclusions convincing and believe this is an important contribution to the mammalian regeneration field. I believe it is more than suitable for publication in *Nature Communications* and should be of interest for both general readers as well as readers specifically interested in regeneration and senescence biology.

Gustavo Tiscornia

Reviewer #4:

Remarks to the Author:

Reviewer 4

I am afraid that my most important requests have not been taken into account;

Either when the authors say they "are developing labelling strategies and should be better equipped to address this question in future manuscripts". This excuse is futile: it is precisely to publish in *Nat Com* such experiments are needed.

When the authors seem to comply to my request to use a more rigorous sensor to compare H₂O₂ levels in fibroblasts of different origins, they use the HyPer probe inappropriately, apparently ignoring that its only rigorous use is through a ratiometric analysis of two excitation wavelengths, not just measuring the green fluorescence!

Important references were missing, which I brought to the authors' attention. Even though the rebuttal letter claims that the authors "appreciate the additional references, with which we are familiar. We have included most at the suggestion of the reviewer where appropriate.", none of the missing references was in fact included. Whether it comes from carelessness, or from a belief that citing relevant previous papers is not appropriate, this inconsistency is a pity.

Consensus Response to Additional Reviewer points

Consensus Response Final Reviewer Comments and Editorial Comments

Reviewer #1 (Remarks to the Author):

The authors have addressed my main concerns in a satisfactory manner.

Reviewer #2 (Remarks to the Author):

The authors have done a tremendous job updating the focus of the paper to center more on their very interesting findings that are specific to reactive oxygen species (ROS) and providing additional supporting evidence around ROS. This work importantly contributes to the field and should be published. Below are minor points that the authors can consider as they finalize the manuscript for publication.

R2.1. *The authors have updated the language and conclusions surrounding results sections 1 and 2. About the sentence “These data provide evidence that increased proliferative ability under physiological oxygen concentrations is associated with regenerative ability in some mammals”. The word association may appear too strong given that based on the proliferation power characterizing these 4 species a chi-squared test will not be able to provide significant evidence that increased proliferative ability is associated with regenerative ability. Perhaps if the authors wish to support this idea based on existing literature, then this statement could be made in the introduction or discussion sections. Instead commenting the results as two contrasting properties would be appropriate: one phenotype that segregates (ROS response), and another that does not (proliferative capacity). Alternatively, the authors could indicate that proliferative capacity seems to be species specific while response to ROS is regeneration-associated.*

We thank the reviewer for pointing this out and have included a sentence as per their suggestion in the discussion.

R2.2. *With respect to the conclusion “Our cross-species comparison reflects cells exhibiting species-specific growth parameters.” The cross-species comparisons reflect multiple variables as indicated by the authors, one of which is the species from which the cells are derived. Another of which is the age of the animals they are derived from. In order to better clarify the author’s idea to the readers that the dominating difference is the species, the authors could cite in the manuscript the references they provided in the rebuttal: Parrinello et al., 2003, Nature Cell Biology and Patrick et al., 2016. Aging (Patrick et al 2016 is cited already, but for a different purpose).*

We appreciate this suggestion. However, in our revision, this language was removed from the manuscript.

R2.3. *While ruling out the possible contribution of structural features of the enzyme in its heightened activity in regenerated animal, the authors have suggested an interesting alternate explanation of how the heightened GPx activity in response to stress is achieved. NAC experiments suggest that an altered kinetics of the GSH pathway or larger stores can also lead to protection against H₂O₂. Adding in the discussion about future directions to figure out the kinetics of GSH metabolism/glutathione stores in regenerating versus non-regenerating mammals may provide further satisfaction to the readers.*

R2.4. *The experiments about GPx and NAC provide insight into how the resistance to senescence is achieved. They open the door for future experiments beyond this paper where the authors can test this by directly comparing the glutathione stores/GSH kinetics in regenerating versus non-regenerating mammals. It may be worth to expand on this in the discussion.*

This is a great suggestion and we now include a sentence in the discussion to highlight the points raised in these two comments.

The new sets of experiments on the structural and functional integrity of mitochondria bring us closer to understanding the intracellular and metabolic differences between regenerating and non-regenerating mammals. Congratulations to the authors on the efforts and the results that more clearly the mechanistic differences between the two groups.

Reviewer #3 (Remarks to the Author):

Saxena et al. present a revised manuscript focusing on characterization of intrinsic differences between fibroblasts obtained from two species capable of regeneration (Acomys and rabbit) in comparison to two species known to heal by fibrotic scarring (rat and mouse). They characterize fibroblasts in terms of their proliferative capacity in relation to oxygen tension and resistance to senescence. They extend this comparison to situations of oxidative and gamma radiation induced stress. They complete their characterization by comparing mitochondrial associated phenotypes and mechanisms of detoxification of free radicals, identifying resistance to senescence as a key characteristic of fibroblasts in regenerating mammals. This revised manuscript is significantly strengthened in comparison to their initial submission. They have changed the structure of the text, presented additional data and for the most part, have addressed my concerns to their first manuscript. Their work is very solid, quantitative and statistically sound. Their discussion is well written and integrates their results well into the literature. I find their conclusions convincing and believe this is an important contribution to the mammalian regeneration field. I believe it is more than suitable for publication in Nature Communications and should be of interest for both general readers as well as readers specifically interested in regeneration and senescence biology.

Reviewer #4 (Remarks to the Author):

R4.1. *I am afraid that my most important requests have not been taken into account; Either when the authors say they "are developing labelling strategies and should be better equipped to address this question in future manuscripts". This excuse is futile: it is precisely to publish in Nat Com such experiments are needed.*

We already responded to their previous comment as follows:

Previous R4.1. *Throughout the manuscript, fibroblast populations are treated as if they were homogenous. Dermis fibroblasts (for instance) are indeed heterogenous, and this is particularly interesting in the context of regeneration, though not taken into consideration here, which is unfortunate.*

We appreciate the reviewer's comment here. As we replied to reviewer 3 (R3.10), we embrace that we are working with a heterogeneous population of cells. In fact, we tried to make this clear in our results section and were clear to state that we isolated connective tissue cells from the ear pinna. However, in light of this comment, we have refined our wording to indicate that we isolated a heterogeneous population of ear pinna connective tissue fibroblasts. Our use of vimentin staining confirms this. We also agree with the reviewer that considering subpopulations of cells in the context of regeneration is interesting. Alongside single-cell data, we are developing labeling strategies and should be better equipped to address this question in future manuscripts.

To reiterate, we do not claim in our manuscript to work with a homogeneous population of cells. We are explicit to state that they are heterogeneous. We do not have transgenics in our system and thus are unable to track/label subpopulations of fibroblasts. As the reviewer is no doubt aware, natural fibroblast heterogeneity and its role during complex tissue regeneration, is poorly understood. Even in mice where transgenics are mature, this question is only just now being teased apart during wound healing. The question of fibroblast heterogeneity is not the focus of our manuscript and thus we believe it is beyond the scope of our study. This does not discount its importance or our interest, but to rigorously pursue this avenue of questioning is material for an entirely new study.

R4.2. *When the authors seems to comply to my request to use a more rigorous sensor to compare H2O2 levels in fibroblasts of different origins, they use the HyPer probe inappropriately, apparently ignoring that its only rigorous use is through a ratiometric analysis of two excitations wavelength, not just measuring the green fluorescence!*

We appreciate the reviewer's passion but respectfully disagree with him/her on this point. While ratiometric measurement using two wavelength excitation with the HyPer probes is often used (and may even be preferred for a level of precision deployed in certain studies), single wavelength monitoring using the HyPer probe is appropriate when cell movements are minimal and sensor expression levels between cells are minimized or taken into account (Belousov et al., 2006, Markvicheva et al., 2008; Bilan and Belousov 2016; Tong et al., 2019). Briefly, to address limitations associated with hydrogen peroxide detection via indirect monitoring, Vsevolod V.

Belousov created a genetic H₂O₂ sensor named HyPer. This sensor consists of yellow fluorescent protein (cpYFP) fused to regulatory domains of OxyR, a peroxide sensitive protein in bacteria. Upon reaction of OxyR with H₂O₂, the intramolecular disulfide bond was formed which results in the conformational change and modifies the cpYFP fluorescence intensity. Using this probe, Belousov and colleagues first demonstrated that a ratiometric analysis of excitation spectra at 420 and 500nm could accurately report intracellular hydrogen peroxide levels. They reported that a ratiometric approach allowed one to avoid potential artifacts associated with significant cell movement or highly variable sensor expression levels. However, in the original publication and subsequent papers, researchers have reported that single wavelength monitoring using these probes could also accurately capture intracellular H₂O₂ dynamics (Belousov et al., 2006; Bilan and Belousov 2016; Markvicheva et al., 2008; Tong et al., 2019). In our current studies, cells exhibited little to no movement over the course of live imaging and we controlled for variable sensor levels, details which we report in our methods section. Instead of blindly selecting cells from across our detection wells, we selected cells from across all wells/species/lines that exhibited similar levels of baseline fluorescence to control for variable sensor expression within and across species. Our time series measurements were calculated as a ratio compared to baseline to provide an accurate, quantitative measure of H₂O₂ dynamics in each cell at a level of precision necessary to test our hypotheses. Thus, the methodology we employ in our manuscript using the HyPer probe is not incorrect, as the reviewer states, but instead, supported by work from those who created the HyPer probe and from other labs deploying this sensor. Our data using the qualitative PO1 sensor further supports this approach. In lieu of the reviewer's comments we have (1) expanded the methods section for the HyPer and Image analysis to include a more detailed account of our approach and supporting references and (2) added two sentences to this effect in the results section where we present our findings with the HyPer.

R4.3. Important references were missing, which I brought to the authors' attention. Even though the rebuttal letter claims that the authors "appreciate the additional references, with which we are familiar. We have included most at the suggestion of the reviewer where appropriate.", none of the missing references was in fact included. Whether it comes from carelessness, or from a belief that citing relevant previous papers is not appropriate, this inconsistency is a pity.

We would like to apologize for what was in this case, an oversight in leaving out several references linking hydrogen peroxide to complex tissue regeneration in vertebrates. As stated in our original response, we appreciate the reviewer highlighting papers they feel are relevant to our work and hopefully the reviewer noted that we did include references from other comments (e.g., Sharpless and Sherr 2015). The papers by Han et al., (2014) and Ferreira et al., (2018) should have been included in the discussion as they are directly relevant to the work presented in our study. We not include these references in the discussion on page 21.

We did not cite the Labit et al., (2018) paper or the Chen et al., (2016) because we felt they are not appropriate as references. The Labit paper uses two models to explore how opioid signaling affects the injury response of two tissues, one of which is not a model of complex tissue

regeneration (MRL fat restoration) and the other a zebrafish tailfin model. Data presented in that manuscript for zebrafish tailfin trying to link ROS and regeneration are correlative and inconclusive (in our expert opinion). The paper does not do one experiment to mechanistically demonstrate how the inhibitors act specifically, or how opioid signaling regulates fat production or whether proliferation is involved, etc. Th paper heavily cites another paper out of one of the senior author's labs (Vriz Lab – Gauron et al., 2013 Sci. Reports) which is a more relevant paper and one we do cite in our manuscript demonstrating a mechanistic link between ROS and complex tissue regeneration.

The Chen et al., (2016) paper, while a nice piece of work, looks at the requirement for ROS during epithelial restoration following exfoliation in zebrafish skin. Here the model is not comparable and citing that paper does not robustly support what is being discussed. For these reasons we do not cite these papers.